# Polyhedral Complex Derivation from Piecewise Trilinear Networks

**Jin-Hwa Kim**
NAVER AI Lab & SNU AIIS
Republic of Korea
j1nhwa.kim@navercorp.com

## Abstract

Recent advancements in visualizing deep neural networks provide insights into their structures and mesh extraction from Continuous Piecewise Affine (CPWA) functions. Meanwhile, developments in neural surface representation learning incorporate non-linear positional encoding, addressing issues like spectral bias; however, this poses challenges in applying mesh extraction techniques based on CPWA functions. Focusing on trilinear interpolating methods as positional encoding, we present theoretical insights and an analytical mesh extraction, showing the transformation of hypersurfaces to flat planes within the trilinear region under the eikonal constraint. Moreover, we introduce a method for approximating intersecting points among three hypersurfaces contributing to broader applications. We empirically validate correctness and parsimony through chamfer distance and efficiency, and angular distance, while examining the correlation between the eikonal loss and the planarity of the hypersurfaces. The code is available at https://github.com/naver-ai/tropical-nerf.pytorch.

## 1 Introduction

Recent advancements in visualizing deep neural networks [1–4] significantly contribute to understanding their intricate structures. This progress provides valuable insights into the expressivity, robustness, training methodologies, and distinctive geometry of neural networks. By leveraging the inherent piecewise linearity in the Continuous Piecewise Affine (CPWA) functions, *e.g.*, ReLU neural networks, each region of the input space is represented as a convex polyhedron. The assembly of these sets constructs a polyhedral complex that delineates the decision boundaries of neural networks [5].

Building upon these breakthroughs, we explore the exciting prospect of analytically extracting a mesh representation from neural implicit surface networks [6–8]. The extraction of a mesh not only offers a precise visualization and characterization of the network's geometry but also does so efficiently through the utilization of vertices and faces, in contrast to sampling-based methods [9, 10]. The sampling-based methods are limited by their black-box nature, which can obscure the underlying structure and lead to inefficiencies in capturing fine geometric details.

These networks typically learn a signed distance function (SDF) by utilizing ReLU neural networks. However, recent approaches incorporate non-linear positional encoding techniques, such as trigonometric functions [11] or trilinear interpolations using hashing [12] or tensor factorization [13], to mitigate issues like spectral bias [14], ensuring fast convergence, and maintaining high-fidelity. Consequently, applying mesh extraction techniques based on CPWA functions becomes challenging.

Focusing on the successful and widely-adopted trilinear interpolating methods, we present novel theoretical insights and a practical methodology for precise mesh extraction. Following a novel review

38th Conference on Neural Information Processing Systems (NeurIPS 2024).

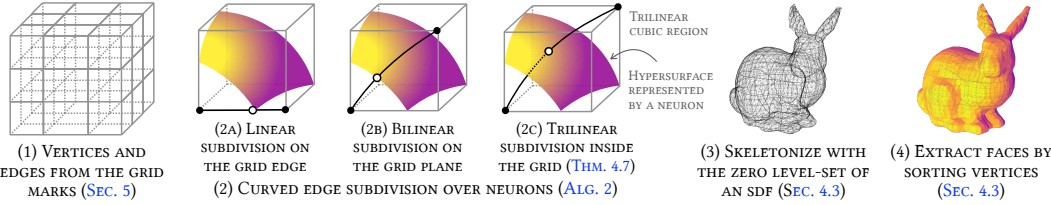

| (1) Vertices and edges from the grid marks (Sec. 5) | (2a) Linear subdivision on the grid edge | (2b) Bilinear subdivision on the grid plane | (2c) Trilinear subdivision inside the grid (Thm. 4.7) | (3) Skeletonize with the zero level-set of an SDF (Sec. 4.3) | (4) Extract faces by sorting vertices (Sec. 4.3) |

(2) Curved edge subdivision over neurons (Alg. 2)

Figure 1: The mesh is analytically extracted from piecewise trilinear networks, comprising both HashGrid [12] and ReLU neural networks, that have been trained to learn a signed distance function with the eikonal loss. We start with initial vertices and edges defined using the grid marks (1), and its linear subdivision (2a); however, intersecting polygons (*ref.* Section 3.2) need bilinear subdivision (2b) and consequentially trilinear subdivision (2c). Note that the linear and bilinear subdivisions are specific instances of trilinear subdivisions with straightforward solutions.

of the *edge subdivision* method [4], an efficient mesh extraction technique for CPWA functions, within the framework of tropical geometry (Section 3), we demonstrate that ReLU neural networks incorporating the trilinear module are piecewise trilinear networks (Definition 4.1). Then, we establish that the hypersurface within the trilinear region becomes a plane under the eikonal [15] constraint (Theorem 4.5), a common practice during the training of SDF.

Building upon this observation, we propose a method for approximating the determination of intersecting points among three hypersurfaces. Given the inherent curvature of these surfaces, achieving an exact solution is infeasible. In our approach, we utilize two hypersurfaces and a diagonal plane for a more feasible and precise approximation (Theorem 4.7). Additionally, we introduce a methodology for arranging vertices composed of faces to implicitly represent normal vectors. We experimentally confirm accuracy and simplicity using chamfer distance, efficiency, and angular distance, simultaneously exploring the relationship between the eikonal loss and the planarity of the hypersurface. In Figure 1, we present the analytically generated skeleton and normal map produced by our method using piecewise trilinear networks composed of HashGrid and ReLU neural networks. For a more detailed view, please refer to Figure 10 in Appendix.

Our contributions are summarized as follows:

1. We present novel theoretical insights and a practical methodology for precise mesh extraction, from the piecewise trilinear networks using the HashGrid.

2. We provide a theoretical analysis of the piecewise trilinear networks under the eikonal constraint revealing that, within a trilinear region, a hypersurface transforms into a plane.

3. We validate its correctness and parsimony through chamfer distance and efficiency, and angular distance, showing the correlation between the eikonal loss and hypersurface planarity.

## 2   Related work

**Mesh extraction in deep neural networks.**   A conventional black-box approach is to assess a given function at sampled points to derive the complex of the function, commonly through marching cubes [9] or marching tetrahedra [10, 16]. Despite the inclusion of mesh optimization techniques [17] in modern mesh packages, this exhaustive method often generates redundant complexities due to discretization, increasing the computational load by using unnecessary mesh elements (*e.g.*, fragmented planars). Aiming for exact extraction, initial efforts focused on identifying linear regions [18] and employing *Analytical Marching* [2] to exactly reconstruct the zero-level isosurface. Contemporary approaches involve region subdivision, wherein the regions of the complex are progressively subdivided from neuron to neuron and layer to layer, allowing for the efficient calculation of the exponentially increasing linear regions [4, 19–21]. Nevertheless, many previous studies have explored CPWA functions consisting of fully connected layers and ReLU activation, primarily owing to the mathematical simplicity of hyperplanes defining linear regions. However, this restricts their applications in emerging domains of deep implicit surface learning that diverge from CPWA functions.

**Positional encoding.** The spectral bias of a multi-layer perceptron (MLP) hinders effectively learning high frequencies, both theoretically and empirically [14]. To address this limitation, Fourier feature mapping, employing trigonometric projections similar to those used in the Transformer architecture [22], proves successful in representing complex 3D objects and scenes. Furthermore, to address its slow convergence and enhance rendering quality, novel positional encoding methods using trilinear interpolations, *e.g.*, HashGrid [12] or TensoRF [13], are introduced. Both techniques generate positional features through trilinear interpolation among the eight nearest corner features derived from pre-defined 3D grids. These features are obtained by hashing from multi-resolution hash tables or factorized representations of a feature space, respectively. However, these approaches cause the function represented by neural networks to deviate from CPWA functions, as trilinear interpolation is not an affine transformation. Consequently, the complex regions are divided by intricate hypersurfaces.

**Eikonal equation and SDF.** An eikonal [1] equation is a non-linear first-order partial differential equation governing the propagation of wavefronts. Notably, this finds application in the regularization of a signed distance function (SDF) in the context of neural surface modeling [6, 8, 23]. If $\Omega$ represents a subset of the space $\mathcal{X}$ equipped with the metric $d$, the SDF $f(\boldsymbol{x})$ is defined as follows:

$$f(\boldsymbol{x}) = \begin{cases} -d(\boldsymbol{x}, \partial\Omega) & \text{if } \boldsymbol{x} \in \Omega \\ d(\boldsymbol{x}, \partial\Omega) & \text{if } \boldsymbol{x} \in \Omega^{\complement} \end{cases} \tag{1}$$

where $\partial\Omega$ denotes the boundary of $\Omega$. The metric $d$ is defined (with a slight notational abuse) as:

$$d(\boldsymbol{x}, \partial\Omega) := \inf_{\boldsymbol{y} \in \partial\Omega} d(\boldsymbol{x}, \boldsymbol{y}) \tag{2}$$

where $d$ represents the shortest distance from $\boldsymbol{x}$ to $\partial\Omega$ in the Euclidean space of $\mathcal{X}$. In the Euclidean space with a piecewise smooth boundary, the SDF exhibits differentiability almost everywhere, and its gradient adheres to *the eikonal equation*. Specifically, for points $\boldsymbol{x}$ on the boundary, $\nabla f(\boldsymbol{x}) = N(\boldsymbol{x})$ where the Jacobian of $f$ represents the outward normal vector, or equivalently, $|\nabla f(\boldsymbol{x})| = 1$.

## 3 Preliminaries

We present an overview of tropical geometry (Section 3.1) to provide a formal definition of the tropical hypersurface and the tropical algebra of neural networks. Then, we discuss the edge subdivision algorithm [4] for extracting the tropical hypersurface (Section 3.2). We supplement Appendix A.1 for less familiar readers. Later, we extend this idea to include considerations for trilinear interpolation in Section 4. You may choose to skip these sections if you are already acquainted with these concepts.

### 3.1 Tropical geometry

Tropical [2] geometry [25, 26] describes polynomials and their geometric characteristics. As a skeletonized version of algebraic geometry, tropical geometry represents piecewise linear meshes using *tropical semiring* and *tropical polynomial* (*ref.* Appendix A.1). The set of points where a tropical polynomial $f$ is non-differentiable, due to tropical sum $\oplus := \max(\cdot)$, is called *tropical hypersurface*:

**Definition 3.1** (Tropical hypersurface). The tropical hypersurface of a tropical polynomial $f(\boldsymbol{x}) = c_1 \boldsymbol{x}^{\alpha_1} \oplus \cdots \oplus c_r \boldsymbol{x}^{\alpha_r}$ is defined as:

$$V(f) = \{\boldsymbol{x} \in \mathbb{R}^d : c_i \boldsymbol{x}^{\alpha_i} = c_j \boldsymbol{x}^{\alpha_j} = f(\boldsymbol{x}) \quad \text{for some} \quad \alpha_i \neq \alpha_j\}$$

where the tropical monomial $c_i \boldsymbol{x}^{\alpha_i}$ with $d$-variate $\boldsymbol{x}$ is defined as $c_i \odot x_1^{a_{i,1}} \odot \cdots \odot x_d^{a_{i,d}}$ with $d$-variate $\alpha_i = (a_{i,1}, \ldots, a_{i,d})$ and *tropical product* $\odot := +$.

**Definition 3.2** (ReLU neural networks). Assuming input and output dimensions are consistent, $L$-layer neural networks with ReLU activation are a composition of functions defined as:

$$\nu^{(L)} = \rho^{(L)} \circ \sigma^{(L-1)} \circ \rho^{(L-1)} \cdots \sigma^{(1)} \circ \rho^{(1)} \tag{3}$$

where $\sigma(\boldsymbol{x}) = \max(\boldsymbol{x}, 0)$ and $\rho^{(i)}(\boldsymbol{x}) = \boldsymbol{W}^{(i)}\boldsymbol{x} + \boldsymbol{b}^{(i)}$.

---

[1]The *eikonal* is derived from the Greek, meaning image or icon.

[2]The *tropical* is named in honor of Hungarian-born Brazilian computer scientist Imre Simon [24].

Although the neural networks $\nu$ is generally non-convex, it is known that the difference of two tropical *signomials* (allowing a real-valued $\alpha$ in a tropical polynomial) can represent $\nu$ [27] (*ref.* Appendix A.3). Given this observation, we proceed to examine the following proposition:

**Proposition 3.3** (Decision boundary of neural networks). *The decision boundary for L-layer neural networks with ReLU activation $\nu^{(L)}(\boldsymbol{x}) = 0$ is a subset of or equals with the region where two tropical monomials, or two arguments of* max*, equal (Definition 3.1), as follows:*

$$\mathcal{B} \subseteq \left\{ \boldsymbol{x} : \nu_j^{(i)}(\boldsymbol{x}) = 0, \ i \in (1, \ldots, L), \ j \in (1, \ldots, H) \text{ if } i \neq L \text{ else } (1) \right\}$$

*where the subscript $j$ denotes the $j$-th neuron, assuming the hidden size of neural networks is $H$, while the final $L$-th layer has a single output for rather discussions as an SDF (Definition 4.3). Remind that tropical operations satisfy the usual laws of arithmetic for recursive consideration.*

The piecewise hyperplanes corresponding to $\nu_j^{(i)}$ are the *candidates* shaping the decision boundary. It enables for-loop iterations over the neurons across layers, as in Algorithm 2, not visiting every exponentially growing linear region [18, 27]. The equality in Proposition 3.3 generally does not hold, but we can filter them by checking if $\nu_1^{(L)}(\boldsymbol{x}) = 0$. For more details, please refer to Berzins [4]. We note that our formal explanation through tropical geometry is unprecedented in the previous work [4].

## 3.2 Edge subdivision for tropical geometry

Polyhedral [3] complex derivation from a multitude of linear regions is inefficient and may be infeasible in some cases. Rather, Berzins [4] argue that tracking vertices and edges of the decision boundary sequentially considering the polynomial number of hyperplanes is particularly efficient, even enabling parallel tensor computations for edge subdivisions.

Initially, we start with a unit cube of eight vertices and a dozen edges, denoted by $\mathcal{V}$ and $\mathcal{E}$, respectively. For each piecewise linear, or *folded* in their term, hyperplane, we find the intersection points of each one of the edges and the hyperplane (if any) to add to $\mathcal{V}$. The divided edges by the intersection are also added to $\mathcal{E}$. Note that we also find new edges of polygons, the intersections of convex polyhedra representing corresponding linear regions [5], and the folded hyperplane. Repeating these for all folded hyperplanes, before selecting all vertices $\boldsymbol{x}^\star$ where $\nu(\boldsymbol{x}^\star) = 0$. Then, the valid edges connecting those vertices would represent the decision boundary.

To elaborate with details, the order of subdividing is invariant as stated in Proposition 3.3; however, we must perform the subdivision with all hyperplanes in the previous layers in advance if we want to keep the current set of edges not crossing linear regions. One critical advantage of *this principle* is that we can find the intersection point by evaluating the output ratio of the two vertices of an edge, $\nu(\boldsymbol{x}_0)$ and $\nu(\boldsymbol{x}_1)$, which are proportional to the distances to the hyperplane by Thales's theorem [28]. The intersection point would simply be $\hat{\boldsymbol{x}}_{0,1} = (1 - w)\boldsymbol{x}_0 + w\boldsymbol{x}_1$ where $w = |d_0|/|d_0 - d_1|$ and $d_k = \nu_j^{(l)}(\boldsymbol{x}_k)$, upon the fact that $d_0 \cdot d_1 < 0$ if the hyperplane divides the edge.

Updating edges needs two steps: dividing the edge into two new edges and adding new edges of intersectional polygons by a hyperplane for convex polyhedra representing linear regions. The latter is a tricky part that we need to find every pair of two vertices among $\{\hat{\boldsymbol{x}}_{\cdot,\cdot}\}$, both within the same linear region *and* on the two common hyperplanes, obviously including the current hyperplane. They argue that the *sign-vectors* for the preactivations provide an efficient way to find them.

**Definition 3.4** (Sign-vectors). For the current hyperplane specified by the $i^*$-th layer and its $j^*$-th preactivation, we define the sign-vectors with a small positive constant $\epsilon$:

$$\gamma(\boldsymbol{x}_k)_{\phi(i,j)} = \begin{cases} +1 & \text{if } \nu_j^{(i)}(\boldsymbol{x}_k) > +\epsilon \\ 0 & \text{if } |\nu_j^{(i)}(\boldsymbol{x}_k)| \leq +\epsilon \\ -1 & \text{if } \nu_j^{(i)}(\boldsymbol{x}_k) < -\epsilon \end{cases} \tag{4}$$

for all $i$ and $j$ where $\phi(i,j) \leq \phi(i^*, j^*)$, and $\phi(i,j) \in \{0\} \cup \mathbb{N}$ is an ascending indexing function from lower layers, handling arbitrary hidden sizes of networks to vectorize a signed matrix.

---

[3]In geometry, a polyhedron is a three-dimensional shape with polygonal faces, having straight edges and sharp vertices.

Notice that a hyperplane divides a space into two half-spaces, two linear regions. Collectively, if the sign-vectors for the $i$-th layer and its $j$-th preactivation of two vertices have the same values except for zeros, which are wild cards for matching, we can say that the two vertices are within the same linear region in the current step $(i^*, j^*)$. For the second, the sign-vector has at least three zeros specifying a point by at least three hyperplanes. If two zeros are at the same indices in the two sign-vectors, the two vertices are on the same two hyperplanes, forming an edge. Notice that our principle asserts edges should be within linear regions.

Berzins [4] argue that the edge subdivision provides the optimal time and memory complexities of $\mathcal{O}(|\mathcal{V}|)$, linear in the number of vertices, while it can be efficiently computed in parallel. According to Hanin and Rolnick [20], the number of linear regions is known for $o(N^L/L!)$ where $N$ is the total number of neurons, or $\phi(L, 1)$ in our notation.

## 4 Method

### 4.1 Piecewise trilinear networks

**Definition 4.1** (Trilinear interpolation). Let a unit cube be in the $D$-dimensional space, where its corners are represented by $\boldsymbol{H} \in \mathbb{R}^{2^D \times F}$ where $F$ is a corner-feature dimension. Given weights $\boldsymbol{w} \in [0, 1]^D$, the trilinear interpolation function $\psi$ is defined as:

$$\psi(\boldsymbol{w}; \boldsymbol{H}) = \sum_{i=0}^{2^D-1} \omega(i, \boldsymbol{w}) \cdot \boldsymbol{H}_i \quad \in \mathbb{R}^F \tag{5}$$

where the interpolating weight $\omega(i, \boldsymbol{w}) \in \mathbb{R}$ is defined using a left-aligned and zero-trailing binarization function $\mathrm{B} \in \{0, 1\}^D$ (*ref.* Appendix B.1) as follows:

$$\omega(i, \boldsymbol{w}) = \prod_{j=1}^{D} (1 - \mathrm{B}(i)_j)(1 - \boldsymbol{w}_j) + \mathrm{B}(i)_j \boldsymbol{w}_j \tag{6}$$

which is the volume of the opposite subsection of the hypercube divided by the weight point $\boldsymbol{w}$. For a general hypercube, scaling the weight for a unit hypercube gets the same result. Going forward, we assume $D=3$ for trilinear interpolation without explicitly stating it otherwise.

**Lemma 4.2** (Nested trilinear interpolation). *Let a nested cube be inside of a unit cube, where its positions of the eight corners deviate $-\boldsymbol{a}_j$ or $+\boldsymbol{b}_j$, such that $\boldsymbol{a}_j, \boldsymbol{b}_j \geq 0$, from $\boldsymbol{w}_j$ for each dimension $j$, using the notations of Definition 4.1. The eight corners of the nested cube are the trilinear interpolations of the eight corners of the unit cube. Then, the trilinear interpolation with the unit cube and the nested cube for $\boldsymbol{w}$ is identical.*

*Proof.* Notice that trilinear interpolation is linear for each dimension to prove it. The detailed proof can be found in Lemma D.1. □

**Definition 4.3** (Piecewise trilinear networks). Let a positional encoding module be $\tau : \mathbb{R}^D \to \mathbb{R}^F$ using the trilinear interpolation of spatially nearby learnable vectors on a three-dimensional grid, *e.g.*, HashGrid [12] or TensoRF [13], where $\tau$ is an instance of $\psi$ in Definition 4.1. Usually, they transform the input coordinate $\boldsymbol{x}$ to $\boldsymbol{w}$ as a relative position inside a unit grid in a corresponding resolution. $\boldsymbol{H}$ is a learnable parameter specified in the corresponding method. Then, we define trilinear neural networks $\tilde{\nu}$, by prepending $\tau$ to the neural networks $\nu$ from Definition 3.2. Here, the input and output dimensions of $\nu^{(L)}$ are $F$ and 1 (an SDF distance), respectively.

$$\tilde{\nu}^{(L)} = \nu^{(L)} \circ \tau \tag{7}$$

Note that $\tau$ is *gridwise* trilinear. Using Lemma 4.2, $\tilde{\nu}$ is trilinear within the composite intersections of linear regions of $\nu$ and trilinear regions of $\tau$. (We discuss how to access *virtual* cubic corner features where trilinear regions are not cubic in Section 5.) So, we regard $\tilde{\nu}$ as *piecewise* trilinear.

## 4.2 Curved edge subdivision in a trilinear space

In a trilinear space, $\tau$ projects a line to a curve except the lines on the grid. Notably, the diagonal line $\boldsymbol{t} = (t, t, t)^\mathsf{T}$ where $t \in [0, 1]$ is projected to a cubic Bézier curve (*ref.* Proposition D.2). We aim to generalize for the *curved* edge subdivision for *hypersurfaces*.

**Lemma 4.4** (Curved edge of two hypersurfaces). *In a piecewise trilinear region, let an edge $(\boldsymbol{x}_0, \boldsymbol{x}_7)$ be the intersection of two hypersurfaces $\tilde{\nu}_i(\boldsymbol{x}) = \tilde{\nu}_j(\boldsymbol{x}) = 0$, while $\boldsymbol{x}_0$ and $\boldsymbol{x}_7$ are on the two hypersurfaces. Then, the edge is defined as:*

$$\{\boldsymbol{x} : \tau(\boldsymbol{x}) = (1 - t)\tau(\boldsymbol{x}_0) + t\tau(\boldsymbol{x}_7) \ \text{where} \ \boldsymbol{x} \in \mathbb{R}^D \ \text{and} \ t \in \mathbb{R}\}.$$

*Proof.* The detailed proof using the piecewise linearity of $\nu$ is provided in Lemma D.4. $\square$

Notice that it does not decrease the number of variables to find a line solution since the trilinear interpolation $\tau$ *still* forms hypersurfaces making complex cases. Yet, we theoretically demonstrate that the eikonal constraints on $\tilde{\nu}$ render them hyperplanes with linear solutions.

**Theorem 4.5** (Hypersurface and eikonal constraint). *A hypersurface $\tau(\boldsymbol{x}) = 0$ intersects two points $\tau(\boldsymbol{x}_0) = \tau(\boldsymbol{x}_7) = 0$ while $\tau(\boldsymbol{x}_{1...6}) \neq 0$ for the remaining six points. These points form a cube, with $\boldsymbol{x}_0$ and $\boldsymbol{x}_7$ positioned on the diagonal of the cube. The hypersurface satisfies the eikonal constraint $\|\nabla\tau(\boldsymbol{x})\|_2^2 = 1$ for all $\boldsymbol{x} \in [0, 1]^3$. Then, the hypersurface of $\tau(\boldsymbol{x}) = 0$ is a plane.*

**Corollary 4.6** (Affine-transformed hypersurface and eikonal constraint). *Let $\nu_0(\boldsymbol{x})$ and $\nu_1(\boldsymbol{x})$ be two affine transformations defined by $\boldsymbol{W}_i^\mathsf{T}\boldsymbol{x} + \boldsymbol{b}_i$, $i \in \{0, 1\}$. A hypersurface $\nu_0(\tau(\boldsymbol{x})) = 0$ passing two points $\nu_0(\tau(\boldsymbol{x}_0)) = \nu_0(\tau(\boldsymbol{x}_7)) = 0$ while $\nu_0(\tau(\boldsymbol{x}_{1...6})) \neq 0$ satisfies the eikonal constraint $\|\nabla\nu_1(\tau(\boldsymbol{x}))\|_2^2 = 1$ for all $\boldsymbol{x} \in [0, 1]^3$. Then, the hypersurface of $\nu_0(\tau(\boldsymbol{x})) = 0$ is a plane.*

The proofs using its partial derivatives can be found in Theorem D.5 and Corollary D.6.

Since we aim for a practical polyhedral complex derivation that piecewise trilinear networks represent, we replace one of the hypersurfaces forming the curved edge with a diagonal plane in a piecewise trilinear region. We choose this option not only for its mathematical simplicity but also due to the characteristics of trilinear hypersurfaces, as illustrated in Figure 9, Appendix G. Thus, the new vertices lie on at least two hypersurfaces, and the new edges exist on the same hypersurface, while the eikonal constraint minimizes any associated error from this approximation. This error is tolerated by the hyperparameter $\epsilon$ =1e-4 as specified in Definition 3.4 and Section 6. More discussions on this matter can be found in Appendix C.1.

**Theorem 4.7** (Intersection of two hypersurfaces and a diagonal plane). *Let $\tilde{\nu}_0(\boldsymbol{x}) = 0$ and $\tilde{\nu}_1(\boldsymbol{x}) = 0$ be two hypersurfaces passing two points $\boldsymbol{x}_0$ and $\boldsymbol{x}_7$ such that $\tilde{\nu}_0(\boldsymbol{x}_0) = \tilde{\nu}_1(\boldsymbol{x}_0) = 0$ and $\tilde{\nu}_0(\boldsymbol{x}_7) = \tilde{\nu}_1(\boldsymbol{x}_7) = 0$, $\boldsymbol{P}_i := \tilde{\nu}_0(\boldsymbol{x}_i)$ and $\boldsymbol{Q}_i := \tilde{\nu}_1(\boldsymbol{x}_i)$, $\boldsymbol{P}_\alpha = [\boldsymbol{P}_0; \boldsymbol{P}_1; \boldsymbol{P}_4; \boldsymbol{P}_5]$, $\boldsymbol{P}_\beta = [\boldsymbol{P}_2; \boldsymbol{P}_3; \boldsymbol{P}_6; \boldsymbol{P}_7]$, and $\boldsymbol{X} = [(1 - x)^2; \ x(1 - x); \ (1 - x)x; \ x^2]$. Then, $x \in [0, 1]$ of the intersection point of the two hypersurfaces and a diagonal plane of $x = z$ is the solution of the following quartic equation:*

$$\boldsymbol{X}^\mathsf{T}(\boldsymbol{P}_\alpha \boldsymbol{Q}_\beta^\mathsf{T} - \boldsymbol{P}_\beta \boldsymbol{Q}_\alpha^\mathsf{T})\boldsymbol{X} = 0, \quad \text{while} \quad y = \frac{\boldsymbol{X}^\mathsf{T}\boldsymbol{P}_\alpha}{\boldsymbol{X}^\mathsf{T}(\boldsymbol{P}_\alpha - \boldsymbol{P}_\beta)} \quad (\boldsymbol{P}_\alpha \neq \boldsymbol{P}_\beta). \tag{8}$$

*Proof.* Please refer to Theorem D.7 for the detailed proof, which uses linear algebra to rearrange two trilinear equations with two variables to get a solution. $\square$

Note that finding roots of a polynomial is the eigenvalue decomposition of a companion matrix [29], which can be parallelized (*ref.* Lemma D.8). Given that the companion matrix remains small for a quartic equation in $\mathbb{R}^{4 \times 4}$, the overall complexity remains $\mathcal{O}(|\mathcal{V}|)$. For the detailed complexity analysis on the proposed method, please refer to Appendix E.

## 4.3 Skeletonization, faces, and normals

**Skeletonization** selects the vertices such that: $\mathcal{V}^\star = \{\boldsymbol{x} \mid |\tilde{\nu}(\boldsymbol{x})| \leq \epsilon, \boldsymbol{x} \in \mathcal{V}\}$, and the edges $\mathcal{E}^\star$ such that their two vertices are among $\mathcal{V}^\star$. Recall that because a decision boundary is a subset of the *tropical hypersurface* (Definition 3.1 and Proposition 3.3), this step ensures accurate mesh extraction.

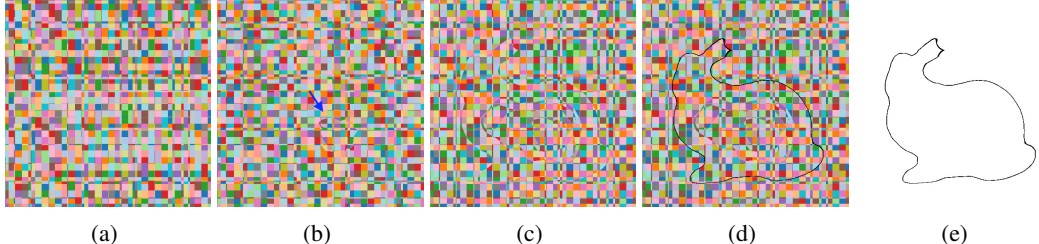

|     |     |     |     |     |
|:---:|:---:|:---:|:---:|:---:|
| (a) | (b) | (c) | (d) | (e) |

Figure 2: Trilinear regions in the xy-plane at $z = 0.04$, identified by the sign-vectors (Definition 3.4), are represented with random colors. (a) Grids described in Section 5 and Algorithm 4. (b) The neurons of the first layer representing folded hypersurfaces (blue arrow). (c) All neurons representing every nonlinear boundary. (d) Select all zero-set vertices and edges. (e) Skeletonized as in Section 4.3.

**Faces** are formed by edges lying on the same plane and sharing a common region. This is identified by evaluating the sign-vectors (*ref.* Definition 3.4 and Section 5). Note that faces are not inherently triangular; however, if needed, they can be triangulated via *triangularization* for further analysis.

**Normals** are conventionally described by the order of the vertices as each face has ambiguity with two opposite directions. Let $\boldsymbol{x}_{0,1,2}$ be the vertices forming a triangular face. The normal is defined using cross-product as $\boldsymbol{n} = (\boldsymbol{x}_1 - \boldsymbol{x}_0) \times (\boldsymbol{x}_2 - \boldsymbol{x}_0)$ where the normal is orthogonal to both vectors, with a direction given by the right-hand rule. Please refer to Appendix B.2 for the details.

## 5  Implementation

We employ the HashGrid [12] for $\tau$, while our discussion on its generalizability is provided in Appendix C.2. We illustrate our method in Algorithm 1, 2, and Figure 2, inspired by Hanin and Rolnick [30] for its visualization of linear regions. We describe the implemental details of the algorithms in the following sections and the corresponding caption of Figure 6.

**Initialization.**  While the original edge subdivision algorithm [4] starts with a unit cube of eight vertices and a dozen edges, we know that the unit cube is subdivided by orthogonal planes consisting of the grid. Let an input coordinate be $\boldsymbol{x} \in [0, 1]^3$, and there is a unit cube such that one of its corners is at the origin. Taking into account multi-resolution grids, we obtain the marks $m_i$, such that the grid planes are $x = m_i$, $y = m_i$, and $z = m_i$, following Algorithm 4 in Appendix. Notice that we carefully consider the offset $s/2$, which prevents the zero derivatives from aligning across all resolution levels (*ref.* Appendix A of Müller et al. [12]). Leveraging the obtained marks, we derive the associated grid vertices and edges, which serve as the initial sets for $\mathcal{V}$ and $\mathcal{E}$, respectively. Please refer to Figure 2a where its multi-resolution grids are visualized using randomized colors.

**Optimizing sign-vectors.**  The number of orthogonal planes may significantly surpass the number of neurons, making it impractical to allocate elements for the sign-vectors (Definition 3.4). To address this, leveraging the orthogonality of the grid planes, we store a plane index along with an indicator of whether the input is on the plane ($\gamma = 0$) or not ($\gamma = 1$).

**Piecewise trilinear region.**  In Theorem 4.7, we need the outputs of corners $\boldsymbol{P}$ and $\boldsymbol{Q}$ in a common piecewise trilinear region; however, some corners may be outside of the region. For this, we replace all ReLU activations using the mask $\boldsymbol{m}$ that: $\boldsymbol{m}_j^{(i)} = (\tilde{\nu}_j^{(i)}(\boldsymbol{x}_0) > \epsilon) \,|\, (\tilde{\nu}_j^{(i)}(\boldsymbol{x}_7) > \epsilon)$, where $\boldsymbol{x}_0$ and $\boldsymbol{x}_7$ are the vertices of an edge of interest, for the calculated cubic corners $\boldsymbol{x}_{0\dots7}$. This implies that $\nu$ exhibits linearity, except for instances where two vertices simultaneously deviate from linearity. For $\boldsymbol{P}$ and $\boldsymbol{Q}$, we select the last two pairs of $i$ and $j$ such that $\gamma_{\phi(i,j)}(\boldsymbol{x}_0) = \gamma_{\phi(i,j)}(\boldsymbol{x}_7) = 0$, indicating two common hypersurfaces passing two points $\boldsymbol{x}_0$ and $\boldsymbol{x}_7$.

## 6  Experiment

**Objective.**  For a given trilinear neural network with the eikonal constraint, our goal is to get a boundary mesh of the zero-set of the SDF. (Note that NeuS [6] showed how to convert the density

| **Algorithm 1** Polyhedral Complex Derivation | **Algorithm 2** Curved Edge Subdivision (CESub) |
|---|---|
| **Input:** Sets of vertices $\mathcal{V}$ and edges $\mathcal{E}$ (from Section 5), # of layer $L$, hidden size $H$, trilinear networks $\tilde{\nu}$
**for** $i = 0$ **to** $L - 1$ **do**
 **for** $j = 0$ **to** $H - 1$ **do**
  $\mathcal{V}, \mathcal{E} = \text{CESub}(\mathcal{V}, \mathcal{E}, \tilde{\nu}, i, j)$
         ▷ Algorithm 2
 **end for**
**end for**
$\mathcal{V}, \mathcal{E} = \text{CESub}(\mathcal{V}, \mathcal{E}, \tilde{\nu}, L, 0)$
$\mathcal{V}^\star, \mathcal{E}^\star = \text{Skeletonize}(\mathcal{V}, \mathcal{E}, \tilde{\nu}, \epsilon)$
         ▷ Section 4.3
$\mathcal{F}^\star = \text{ExtractFaces}(\mathcal{V}^\star, \mathcal{E}^\star)$
         ▷ Section 4.3
**return** $\mathcal{V}^\star, \mathcal{E}^\star, \mathcal{F}^\star$ | **Input:** Vertices $\mathcal{V}$, Edges $\mathcal{E}$, trilinear networks $\tilde{\nu}$, layer index $i$, neuron index $j$
$d = \tilde{\nu}(\mathcal{V}; i, j), \mathcal{V}_{\text{new}} = \emptyset$
**for** $e$ **in** $\mathcal{E}$ **do**
 **if** $d_{e_0} \cdot d_{e_1} < 0$ **then**
  $\boldsymbol{P}, \boldsymbol{Q} = \text{Corners}(e)$   ▷ Section 5
  $\boldsymbol{x}_{\text{new}} = \text{Intersect}(\boldsymbol{P}, \boldsymbol{Q}, e)$ ▷ Theorem 4.7
  $\mathcal{V}_{\text{new}} \leftarrow \mathcal{V}_{\text{new}} + \{\boldsymbol{x}_{\text{new}}\}$
  $e \leftarrow [e_0, \text{Index}(\boldsymbol{x}_{\text{new}})]$
  $\mathcal{E} \leftarrow \mathcal{E} + \{[e_1, \text{Index}(\boldsymbol{x}_{\text{new}})]\}$
 **end if**
**end for**
$\mathcal{V} \leftarrow \mathcal{V} + \mathcal{V}_{\text{new}}$
$\mathcal{E} \leftarrow \mathcal{E} + \text{FindPolygons}(\mathcal{V}_{\text{new}})$  ▷ Section 3.2
**return** $\mathcal{V}, \mathcal{E}$ |

networks of NeRFs to an SDF in the frameworks of volume rendering.) To evaluate this, we utilize marching cubes with excessive sampling to get a pseudo-ground truth mesh to remove Bayes error caused by underfitting. We explicitly specify the samplings in the results.

**Hyperparameters.** We used the number of layers $L$ of 3 and hidden size $H$ of 16 for the networks, and $\epsilon$ of 1e-4 for the sign-vectors (*ref.* Definition 3.4). The weight for the eikonal loss is 1e-2. For HashGrid, the resolution levels of 4, feature size of 2, base resolution $N_{\min}$ of 2, and max resolution $N_{\max}$ of 32 *by default*. $N_{\min}$ and $N_{\max}$ are doubled (x2) or quadrupled (x4) for *Medium* or *Large* settings in Figure 4. We use the official Python package of `tinycudnn` [4] for the HashGrid module.

**Chamfer distance and efficiency.** The chamfer distance is a metric used to evaluate the similarity between two sets of sampled points from two meshes. Let $\mathcal{S}_0$ and $\mathcal{S}_1$ be the two sets of points. Specifically, the bidirectional chamfer distance (CD) is defined as follows:

$$\text{CD}(\mathcal{S}_0, \mathcal{S}_1) = \frac{1}{2}\Big(\overrightarrow{\text{CD}}(\mathcal{S}_0, \mathcal{S}_1) + \overrightarrow{\text{CD}}(\mathcal{S}_1, \mathcal{S}_0)\Big), \ \ \overrightarrow{\text{CD}}(\mathcal{S}_0, \mathcal{S}_1) = \frac{1}{|\mathcal{S}_0|} \sum_{\boldsymbol{x} \in \mathcal{S}_0} \min_{\boldsymbol{y} \in \mathcal{S}_1} \|\boldsymbol{x} - \boldsymbol{y}\|_2^2. \quad (9)$$

We randomly select 100K points on the faces through ray-marching from the origin, directing toward a randomly chosen point on a unit sphere.

Table 1 and Figure 4 show the results for the Standford 3D Scanning repository [31]. Varying the number of samples in marching cubes [9], we plot the baseline with respect to the number of generated vertices. Given that our method extracts vertices from the intersection points of hypersurfaces, it enables parsimonious representations using the lower number of vertices having better CD, summarized by chamfer efficiency (CE) [5] in Table 1, and ours are located in the lower-left region compared with the baselines in Figure 4. In particular, we observe a strong CE in a simple object, *e.g.*, Drill Bit, which achieved a CE of 47.8 versus 19.2 (MC).

Our method inherently holds an advantage over MC in capturing optimal vertices and faces, especially in cases where the target surfaces exhibit planar characteristics. When dealing with curved surfaces, the smaller grid size of the *Large* model in Figure 4 views a curved surface in a smaller interval, and it tends to approach the plane. From the plotting of *Small* to

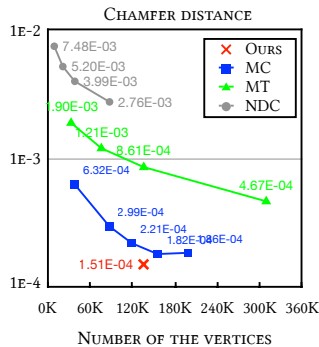

Figure 3: Chamfer distance for the bunny with the *Large* model comparing with MC, MT, and NDC.

---

[4] https://github.com/NVlabs/tiny-cuda-nn
[5] CE is defined as $100/(\sqrt{|\mathcal{V}|} \times \text{CD})$, reflecting the trade-off between the number of vertices and the CD.

Table 1: Chamfer distance (CD) and chamfer efficiency (CE) for the Stanford 3D Scanning repository [31]. The CD ($\times$1e-6) is evaluated using the marching cubes (MC) with $256^3$ grid samples as the reference ground truth for all measurements. Please refer to Table 2 and Table 3 for the full results, and Table 4 for standard deviation and time spent in Appendix (Ours took $0.97 \pm 0.21$ while MC also took within 1 sec for the bunny.)

| | Bunny | | | Dragon | | | Happy Buddha | | | Armadillo | Drill | Lucy |
|---|---|---|---|---|---|---|---|---|---|---|---|---|
| MC # | $\|\mathcal{V}\|\downarrow$ | CD$\downarrow$ | CE$\uparrow$ | $\|\mathcal{V}\|\downarrow$ | CD$\downarrow$ | CE$\uparrow$ | $\|\mathcal{V}\|\downarrow$ | CD$\downarrow$ | CE$\uparrow$ | CE$\uparrow$ | CE$\uparrow$ | CE$\uparrow$ |
| 32 | 1283 | 4106 | 19.0 | 1416 | 6330 | 11.2 | 1032 | 6951 | 13.9 | 11.2 | 7.5 | **24.9** |
| 64 | 5367 | 1371 | 13.6 | 6090 | 1936 | 8.5 | 4362 | 2465 | 9.3 | 8.3 | 19.2 | 16.0 |
| 128 | 21825 | 393 | 11.6 | 25236 | 542 | 7.3 | 18219 | 722 | 7.6 | 6.6 | 17.4 | 10.7 |
| 196 | 49569 | 141 | 14.3 | 57341 | 203 | 8.6 | 41351 | 276 | 8.8 | 7.5 | 15.3 | 11.7 |
| Ours | 4341 | 900 | **25.6** | 5104 | 1243 | **15.8** | 3710 | 1738 | **15.5** | **13.9** | **47.8** | 23.6 |

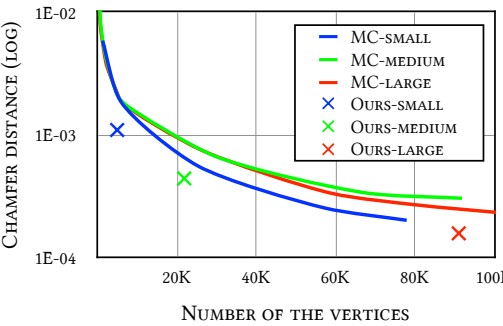

Figure 4: Chamfer distances for the Stanford dragon varying the number of vertices and the model sizes, showing consistent efficiency.

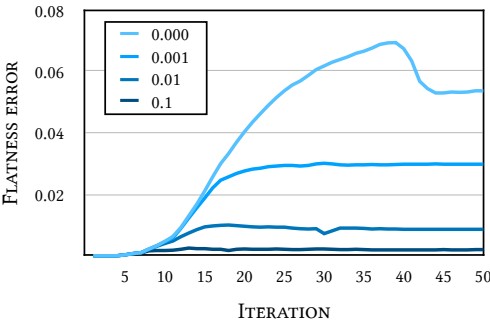

Figure 5: Effect of the weight of eikonal loss on the flatness error in Equation (11). For the plot, we conduct experiments using the unit sphere.

*Large*, our CDs (colored crosses) are decreasing to zero, consistently with better CEs, confirming this speculation. The complete results of the *Large* models and its visualization can be found in Tables 6 to 8 in Appendix F.1 and the right side of Figure 10 in Appendix, respectively. We also compare ours with Marching Tetrahedra (MT) [10, 16] and Neural Dual Contour (NDC) [32] in Figure 3, where our method achieved the lowest chamfer distance and the most efficiency method with respect to the number of vertices. For the rationale behind our selection, please refer to Appendix F.3.

**Angular distance.** The angular distance measures how much the normal vectors deviate from the ground truths. Analogous to the chamfer distance, we sample 100K points on the surface and calculate the normal vectors as described in Section 4.3. The angular distance is defined as follows:

$$\mathrm{AD}(\mathcal{N}_0, \mathcal{N}_1) = \mathbb{E}_i \left[ \frac{180}{\pi} \cdot \cos^{-1} \left( \langle \mathcal{N}_0^{(i)}, \mathcal{N}_1^{(i)} \rangle \right) \right] \quad (10)$$

In Table 5 in Appendix, the angular distances for the Stanford bunny are shown. As we expected, our approach efficiently estimates the faces using a parsimonious number of vertices compared to the sampling-based method. We confirm this trend is consistent across other objects.

**Eikonal constraint for planarity.** To assess the efficacy of the eikonal constraint, which enforces the planarity of hypersurfaces as outlined in Theorem 4.5, we quantify the flatness error associated with the planarity constraints presented in the proof of Theorem D.5. Let $\boldsymbol{x}_0$ and $\boldsymbol{x}_7$ be two vertices consisting of a diagonal edge, while $\boldsymbol{x}_{1\dots6}$ are its remaining cubic corners. We obtain $\tilde{\nu}(\boldsymbol{x}_{1\dots6})$ in a piecewise trilinear region as described in Section 5. This evaluation is conducted as follows:

$$\Delta_{\mathrm{flat}} = \mathbb{E}_{\mathcal{E}} \left[ \frac{1}{4} \left( \|\Sigma_{i \in \{1,2,4\}} \tilde{\nu}(\boldsymbol{x}_i)\|_1 + \|\Sigma_{i \in \{3,5,6\}} \tilde{\nu}(\boldsymbol{x}_i)\|_1 \right) + \frac{1}{6} \left( \sum_{i=1}^{3} \|\Sigma_{j \in \{i, 7-i\}} \tilde{\nu}(\boldsymbol{x}_j)\|_1 \right) \right]. \quad (11)$$

In Figure 5, we empirically validate that the eikonal loss enforces planarity in the hypersurfaces within piecewise trilinear regions, as described in Theorem 4.5. The absence of the eikonal loss results in elevated planarity errors, leading to the construction of an inaccurate mesh.

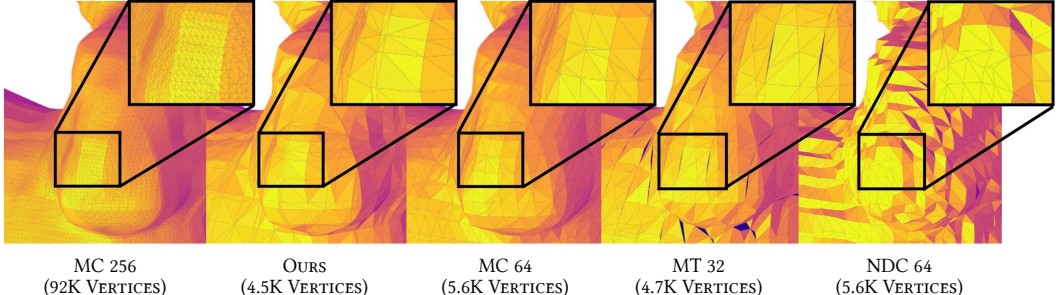

|  MC 256 | Ours | MC 64 | MT 32 | NDC 64 |
| (92K Vertices) | (4.5K Vertices) | (5.6K Vertices) | (4.7K Vertices) | (5.6K Vertices) |

Figure 6: This figure provides a detailed visualization of Figure 10 in the Appendix and its competitors. MC 64, MT 32 (Marching Tetrahedra), and NDC (Neural Dual Contour [32]) suffer over-smoothing or inaccuracy of the actual surface in the Small networks within the nose, whereas our method reflects these details (see the boundaries of a nose) with consistent normals (in colors). MT efficiently demands SDF values by utilizing six tetrahedra within grids to get intermediate vertices. However, it is inefficient in terms of the number of vertices extracted. NDC faces a generalization issue for a zero-shot setting for the given networks, producing unsmooth surfaces.

**Visualization.** We provide the qualitative analyses in Figure 6 comparing with MT [10, 16] and NDC [32], along with MC. While the competitors exhibit over-smoothing or inaccuracies, particularly in the nose region, our method preserves surface details with consistent normals and outperforms in both accuracy and generalization. In addition, we present Figures 10 to 13 in Appendix G encompassing the comparison between the *Small* and *Large* models (Figure 10), visualizing the marching cubes varying the number of samples (Figure 11), detailed comparisons (Figure 12), and the gallery of the Stanford 3D Scanning meshes from the *Large* models (Figure 13).

**Limitations.** Although we implement the algorithm using pre-defined parallel tensor operations in PyTorch [33], achieving time and memory complexity of $\mathcal{O}(|\mathcal{V}|)$ as detailed in Appendix E, our method inherently requires forward passing the intermediate vertices during edge subdivisions iterating over neurons, which depends on previous steps (Algorithm 1). For the Stanford bunny using the *Small* model, the intermediate counts for vertices and edges exceed 11K and 19K, respectively, although they are eventually reduced to 4.3K vertices for face extraction. This overhead is negligible for the *Small* models, taking 0.97 seconds; however, for the *Large* models, the process takes 4.15±0.61 seconds for 140.3±5.6K vertices with our optimized algorithm, compared to 1.31 and 11.88 seconds for marching cubes with 256 and 512 samples per axis, respectively. We note that parallelizing over multiple GPUs would be a reliable option to improve latency. For the discussions on satisfying the eikonal equation in practice and the generalization to other positional encodings using trilinear interpolation, please refer to Appendix C.1 and Appendix C.2, respectively.

## 7 Conclusions

In conclusion, our exploration of novel techniques in visualizing trilinear neural networks unveils its intricate structures and extends mesh extraction applications beyond CPWA functions. Focused on trilinear interpolating methods, our study yields novel theoretical insights and a practical methodology for precise mesh extraction. Notably, Theorem 4.5 shows the transformation of hypersurfaces to planes within the trilinear region under the eikonal constraint. Empirical validation using chamfer distance and efficiency, and angular distance affirms the correctness and parsimony of our methodology. The correlation analysis between eikonal loss and hypersurface planarity confirms the theoretical findings. Our proposed method for approximating intersecting points among three hypersurfaces, coupled with a vertex arrangement methodology, broadens the applications of our techniques. This work establishes a foundation for future research and practical applications in the evolving landscape of deep neural network analysis and toward real-time rendering mesh extraction.

## Acknowledgments

I would like to express my sincere appreciation to my brilliant colleagues, Sangdoo Yun, Dongyoon Han, and Injae Kim, for their contributions to this work. Their constructive feedback and guidance have been instrumental in shaping the work. I also express my sincere thanks to the anonymous reviewers for their help in improving the manuscript. The NAVER Smart Machine Learning (NSML) platform [34] had been used for experiments.

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

# Appendix

## Table of contents

## A   Tropical geometry and tropical algebra of neural networks

In this section, we present an overview of tropical geometry (Appendix A.1) to provide a formal definition of the tropical hypersurface (Appendix A.2). This hypersurface is significant as it serves as the decision boundary for ReLU neural networks optimizing an SDF, resulting in the formation of a polyhedral complex. For this, we delve into the examination of the tropical algebra of neural networks (Appendix A.3). Furthermore, we discuss the edge subdivision algorithm [4] for extracting the tropical hypersurface (Section 3.2), and we extend this discussion to include considerations for trilinear interpolation in Section 4. Given that Appendix A.1 to A.3 offer insights into tropical geometry, illustrating how each neuron contributes to forming the polyhedral complex, you may choose to skip these subsections if you are already acquainted with these concepts.

### A.1   Tropical geometry

Tropical geometry [25, 26] describes polynomials and their geometric characteristics. As a skeletonized version of algebraic geometry, tropical geometry represents piecewise linear meshes using *tropical semiring* and *tropical polynomial*.

The tropical semiring is an algebraic structure that extends real numbers and $-\infty$ with the two operations of *tropical sum* $\oplus$ and *tropical product* $\odot$. Tropical sum and product are conveniently replaced with $\max(\cdot)$ and $+$, respectively. Notice that $-\infty$ is the tropical sum identity, zero is the tropical product identity, and these satisfy associativity, commutativity, and distributivity. For example, a classical polynomial $x^3 + 2xy + y^4$ would represent $\max(3x, \ x + y + 2, \ 4y)$. The

classical polynomial is rewritten for a tropical polynomial by naively replacing $+$ with $\oplus$ and $\cdot$ with $\odot$, respectively. After this, we rewrite by the definitions of tropical operations as $\max(3x,\ x+y+2,\ 4y)$. Roughly speaking, this is the tropicalization of the polynomial, denoted tropical polynomial or simply $f$. Note that this example (ours with max notation) comes from the Wikipedia [6].

Additionally, we present a graphical diagram in Figure 7 illustrating how the tropical polynomial divides the 2D space for clearer understanding. Please see the caption for further details.

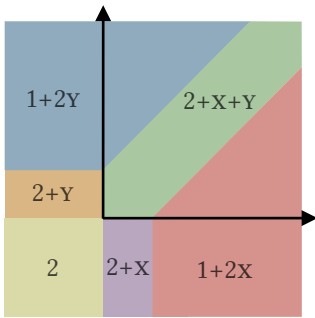

Figure 7: A classical polynomial $x^2 + y^2 + 2xy + 2x + 2y + 2$ turns into a tropical polynomial as $\max(1 + 2x, 1 + 2y, 2 + x + y, 2 + x, 2 + y, 2)$, taking into account the implicit coefficient of 1. We redraw Figure 1 of the tropical curve from Zhang et al. [27] as a region map. Each color indicates a region where each tropical monomial has maximum.

## A.2   Tropical hypersurface

The set of points where a tropical polynomial $f$ is non-differentiable, mainly due to tropical sum, $\max(\cdot)$, is called *tropical hypersurface*, denoted $V(f)$. Note that $V$ is an analogy to the *vanishing set* of a polynomial.

**Definition A.1** (Tropical hypersurface, restated). The tropical hypersurface of a tropical polynomial $f(\boldsymbol{x}) = c_1\boldsymbol{x}^{\alpha_1} \oplus \cdots \oplus c_r\boldsymbol{x}^{\alpha_r}$ is defined as:

$$V(f) = \{\boldsymbol{x} \in \mathbb{R}^d : c_i\boldsymbol{x}^{\alpha_i} = c_j\boldsymbol{x}^{\alpha_j} = f(\boldsymbol{x}) \quad \text{for some} \quad \alpha_i \neq \alpha_j\}$$

where the tropical monomial $c_i\boldsymbol{x}^{\alpha_i}$ with $d$-variate $\boldsymbol{x}$ is defined as $c_i \odot x_1^{a_{i,1}} \odot \cdots \odot x_d^{a_{i,d}}$ with $d$-variate $\alpha_i = (a_{i,1}, \ldots, a_{i,d})$ and *tropical product* $\odot := +$. In other words, this is the set of points where $f(\boldsymbol{x})$ equals two or more monomials in $f$.

## A.3   Tropical algebra of neural networks

**Definition A.2** (ReLU neural networks, restated). Assuming input and output dimensions are consistent, $L$-layer neural networks with ReLU activation are a composition of functions defined as:

$$\nu^{(L)} = \rho^{(L)} \circ \sigma^{(L-1)} \circ \rho^{(L-1)} \cdots \sigma^{(1)} \circ \rho^{(1)} \tag{12}$$

where $\sigma(\boldsymbol{x}) = \max(\boldsymbol{x}, 0)$ and $\rho^{(i)}(\boldsymbol{x}) = \boldsymbol{W}^{(i)}\boldsymbol{x} + \boldsymbol{b}^{(i)}$.

Although the neural networks $\nu$ is generally non-convex, it is known that the difference of two tropical *signomials* can represent $\nu$ [27]. A tropical signomial takes the same form as a tropical polynomial (*ref.* Definition 3.1), but it additionally allows the exponentials $\alpha$ to be real values rather than integers. With tropical signomials $F$, $G$, and $H$,

$$\nu^{(l)}(\boldsymbol{x}) = H^{(l)}(\boldsymbol{x}) - G^{(l)}(\boldsymbol{x}) \tag{13}$$

$$\sigma^{(l)} \circ \nu^{(l)}(\boldsymbol{x}) = F^{(l)}(\boldsymbol{x}) - G^{(l)}(\boldsymbol{x}) \tag{14}$$

---

[6]https://en.wikipedia.org/wiki/Tropical_geometry

where $F^{(l)}$, $G^{(l)}$, and $H^{(l)}$ are as follows:

$$F^{(l)}(\boldsymbol{x}) = \max\left(H^{(l)}(\boldsymbol{x}),\, G^{(l)}(\boldsymbol{x})\right) \tag{15}$$

$$G^{(l)}(\boldsymbol{x}) = \boldsymbol{W}_+^{(l)} G^{(l-1)}(\boldsymbol{x}) + \boldsymbol{W}_-^{(l)} F^{(l-1)}(\boldsymbol{x}) \tag{16}$$

$$H^{(l)}(\boldsymbol{x}) = \boldsymbol{W}_+^{(l)} F^{(l-1)}(\boldsymbol{x}) + \boldsymbol{W}_-^{(l)} G^{(l-1)}(\boldsymbol{x}) + \boldsymbol{b}^{(l)} \tag{17}$$

while $\boldsymbol{W}^{(l)} = \boldsymbol{W}_+^{(l)} - \boldsymbol{W}_-^{(l)}$ where $\boldsymbol{W}_+^{(l)}$ and $\boldsymbol{W}_-^{(l)} \in \mathbb{R}^+$.

**Proposition A.3** (Decision boundary of neural networks, restated). *The decision boundary for L-layer neural networks with ReLU activation $\nu^{(L)}(\boldsymbol{x}) = 0$ is where two tropical monomials, or two arguments of* max*, equal (Definition 3.1), as follows:*

$$\mathcal{B} \subseteq \left\{ \boldsymbol{x} : \nu_j^{(i)}(\boldsymbol{x}) = H_j^{(i)}(\boldsymbol{x}) - G_j^{(i)}(\boldsymbol{x}) = 0,\ i \in (1, \dots, L),\ j \in (1, \dots, H) \text{ if } i \neq L \text{ else } (1) \right\}$$

*where the subscript $j$ denotes the $j$-th neuron, assuming the hidden size of neural networks is $H$, while the L-th layer has a single output for rather discussions as an SDF. Remind that tropical operations satisfy the usual laws of arithmetic for recursive consideration.*

This implies that piecewise hyperplanes corresponding to the preactivations $\{\nu_j^{(i)}\}$ are the candidates shaping the decision boundary. Each linear region has the same pattern of zero masking by $\mathrm{ReLU}(\cdot)$, effectively making it linear as an equation of a plane $\boldsymbol{A}\boldsymbol{x} + \boldsymbol{b} = 0$ for some $\boldsymbol{A}$ and $\boldsymbol{b}$. In turn, if the masking pattern changes as an input is continually changed, it moves to another linear region. From a geometric point of view, this is why the number of linear regions grows polynomially with the hidden size $H$ and exponentially with the number of layers $L$ [18, 27].

# B    Implementation details

## B.1    Zero-trailing binarization function

B is zero-trailing binarization function, consisting of $D = 3$ bits. The zero-trailing means the number in bits is increasing left-aligned. For example, the number one is "100", while the numbers two and four are "010" and "001", respectively. The number three is "110". From the left, each bit indicates x, y, and z axis in our notation. This convention seems to be widely used in the CUDA implementation in C language. Notice that we use the $j$ subscript of $B_j$ to select the axis. Naturally, $i = 0$ and $i = 7$ indicate the opposite two corner points.

## B.2    Sorting polygon vertices and HashGrid marks

Algorithm 3 describes the algorithm in Section 4.3, which are sorting polygon vertices implicitly representing the direction of faces. If the viewing direction aligns opposite to the normal, the order of vertices follows a counter-clockwise arrangement. We determine the vertex order based on the normal derived from the Jacobian of the mean of vertices. We identify the counter-clockwise arrangement of vertices using dot product to compute relative angles $\theta \in [-\pi, \pi]$ and cross product for getting direction. Algorithm 4 describes the algorithm in Section 5, calculating the HashGrid marks from the hyperparameters of a HashGrid.

---

**Algorithm 3** Sorting Polygon Vertices

**Input:** vertices $\boldsymbol{x}_i$, normal $\boldsymbol{n}$, $N = |\{\boldsymbol{x}_i\}|$
$\mu = \sum_i \boldsymbol{x}_i / N$
**for** $i = 0$ **to** $N - 1$ **do**
  $\boldsymbol{v}_i = (\boldsymbol{x}_i - \mu)/\|\boldsymbol{x}_i - \mu\|_2$
  $\boldsymbol{c}_i = \langle \boldsymbol{v}_i,\ \boldsymbol{v}_0 \rangle,\ \ \boldsymbol{d}_i = \langle \boldsymbol{v}_i \times \boldsymbol{v}_0,\ \boldsymbol{n} \rangle$
  $\theta_i = \boldsymbol{c}_i(2 \cdot \mathbf{1}_{\boldsymbol{d}_i \geq 0} - 1) + 2 \cdot \mathbf{1}_{\boldsymbol{d}_i < 0}$
**end for**
**return** $\mathrm{ArgSort}(\{-\theta_i\})$

---

**Algorithm 4** HashGrid Marks

**Input:** Resolution levels $L$, base resolution $N_{\min}$, max resolution $N_{\max}$
$m = \{0,\ 1\}$
$b = \log_2(N_{\max}/N_{\min})/(L - 1)$
**for** $l = 0$ **to** $L - 1$ **do**
  $s = 1/\left(N_{\min} \cdot \exp_2(l \cdot b) - 1\right)$
  $m \leftarrow m + \mathrm{Max}\left(\mathrm{Arange}(-s/2,\ 1,\ s),\ 0\right)$
**end for**
**return** $\mathrm{Sort}\left(\mathrm{Unique}(m)\right)$

---

# C  Further discussions

## C.1  Satisfying the eikonal equation and approximation

Theorem 4.7 handles where the eikonal equation does not make perfect hyperplanes. First, we observed the characteristics of trilinear hypersurfaces as illustrated in Figure 9 in Appendix. Among 64 edge scenarios within a trilinear region, a diagonal plane intersects with the hypersurface in nearly all cases. Moreover, we can take advantage of it since we can eliminate one variable by assuming the curved edge lies on the $x = z$ plane. In this light, Theorem 4.7 used linear algebra to rearrange two trilinear equations to get Equations (71) and (74). As we mentioned in the manuscript, the new vertices lie on at least two hypersurfaces, and the new edges exist on the same hypersurface, while the eikonal constraint minimizes any associated error. This error is tolerated by the hyperparameter $\epsilon$=1e-4 (Section 6) as specified in Definition 3.4.

Figure 5 also suggests empirical evidence. Empirically, the flatness error, derived from the proof of Theorem D.5, is effectively controlled by the learning rate of the eikonal loss. In the experiments, we used 0.01 to give our results, while many other neural implicit surface learning methods were used to exploit a higher learning rate of 0.1 (*e.g.*, Yariv et al. [7]). Note that, in practice, the eikonal loss is only applied to near-surface using clamps (if the target position is obviously far from a surface, ignore it; it is wise since we are concerned with the surfaces, not wanting exact SDFs for all space) for effective learning, which is sufficient to our extraction method.

Moreover, the smaller grid size of the *Large* model (*ref.* Section 6) in Figure 4 views a curved surface in a smaller interval, and it tends to regard it as a plane. From the plotting of *Small* to *Large*, our CD (colored crosses) is decreasing to zero, consistently with better CE, confirming this speculation.

## C.2  Generalization to other positional encodings using trilinear interpolation

We can say that *any encoding using trilinear interpolation (that) we defined can be applied in our method.* Here, the definition is implicitly referred to Definition 4.1 (Trilinear interpolation), where the cubic corners are represented by $\boldsymbol{H} \in \mathbb{R}^{D \times F}$. Since we do not rely on a specific mapping function $f : \mathrm{x} \to (\boldsymbol{H}_0, \boldsymbol{H}_1 \ldots \boldsymbol{H}_7)$, we may apply our extraction method to any encoding using trilinear interpolation that we defined. Specifically, InstantNGP [12] used a hashing function to map a corner point to the arbitrary entry of hash tables, allowing gracious collisions with online adaptivity (refer to their Sec. 3), while TensoRF [13] can be interpreted that it used a mapping function to the summation of the multiplications of two vectors from the vector-matrix factorization. Therefore, if we have the full ranks for $R_1, R_2, R_3$, in Eqn. 3 in Chen et al. [13], among the VM factorization, which can be equivalently represented by the HashGrid with a sufficient number of hash table sizes.

Then, the discussion should be directed toward their encoding characteristics, which impact our extraction method due to their online adaptivity (HashGrid) or low-rank factorization (TensoRF). When we train an SDF, the optimization to global minima is generally difficult to achieve. Instead, previous works on neural implicit surface learning methods resort to optimizing the SDF on near-surface since we are interested in modeling the zero-set surfaces, allowing some errors for the points that are far from the surfaces. Practically, this is done by defining a loss function using clamps and ignoring far-surface points. For this reason, the online adaptivity of HashGrid or low-rank factorization of TensoRF can exploit this nature: focused learning of the surfaces adaptively allocating their learnable parameters to the near-surface, which means there might be enough room for the regularization by the eikonal equation. The investigation of the efficiency of representing the implicit neural surfaces using various positional encodings is seemingly out of the scope of this work since our main interest lies in the theoretical exposition of the mesh extraction from it and setting up a feasible bridge between theory and practice.

# D  Theoretical proofs

**Lemma D.1.** *(Nested trilinear interpolation, restated) Let a nested cube be inside of a unit cube, where its positions of the eight corners deviate $-\boldsymbol{a}_j$ or $+\boldsymbol{b}_j$, such that $\boldsymbol{a}_j, \boldsymbol{b}_j \geq 0$, from $\boldsymbol{w}_j$ for each dimension $j$, using the notations of Definition 4.1. The eight corners of the nested cube are the trilinear interpolations of the eight corners of the unit cube. Then, the trilinear interpolation with the unit cube and the nested cube for $\boldsymbol{w}$ is identical.*

*Proof.* Let a left-aligned and zero-trailing binarization function be $\mathrm{B}(\cdot) \in \{0,1\}^D$ and $D = 3$. The partial derivative of the trilinear interpolation with the nested cube $\hat{\boldsymbol{H}}$ with respect to $\boldsymbol{H}_7$ and $\boldsymbol{H}_0$ would be:

$$\frac{\partial \phi(\boldsymbol{w}; \hat{\boldsymbol{H}})}{\partial \boldsymbol{H}_7} = \sum_{i=0}^{2^D-1} \prod_{j=1}^{D} \frac{(1 - \mathrm{B}(i)_j)\boldsymbol{b}_j(\boldsymbol{w}_j - \boldsymbol{a}_j) + \mathrm{B}(i)_j \boldsymbol{a}_j(\boldsymbol{w}_j + \boldsymbol{b}_j)}{\boldsymbol{a}_j + \boldsymbol{b}_j} \tag{18}$$

$$= \sum_{i=0}^{2^{D-1}-1} \frac{\boldsymbol{b}_D(\boldsymbol{w}_D - \boldsymbol{a}_D) + \boldsymbol{a}_D(\boldsymbol{w}_D + \boldsymbol{b}_D)}{\boldsymbol{a}_D + \boldsymbol{b}_D} \cdot$$

$$\prod_{j=1}^{D-1} \frac{(1 - \mathrm{B}(i)_j)\boldsymbol{b}_j(\boldsymbol{w}_j - \boldsymbol{a}_j) + \mathrm{B}(i)_j \boldsymbol{a}_j(\boldsymbol{w}_j + \boldsymbol{b}_j)}{\boldsymbol{a}_j + \boldsymbol{b}_j} \tag{19}$$

$$= \boldsymbol{w}_D \sum_{i=0}^{2^{D-1}-1} \prod_{j=1}^{D-1} \frac{(1 - \mathrm{B}(i)_j)\boldsymbol{b}_j(\boldsymbol{w}_j - \boldsymbol{a}_j) + \mathrm{B}(i)_j \boldsymbol{a}_j(\boldsymbol{w}_j + \boldsymbol{b}_j)}{\boldsymbol{a}_j + \boldsymbol{b}_j} \tag{20}$$

$$= \prod_{j=1}^{D} \boldsymbol{w}_j = \frac{\partial \phi(\boldsymbol{w}; \boldsymbol{H})}{\partial \boldsymbol{H}_7} \tag{21}$$

and

$$\frac{\partial \phi(\boldsymbol{w}; \hat{\boldsymbol{H}})}{\partial \boldsymbol{H}_0} = \sum_{i=0}^{2^D-1} \prod_{j=1}^{D} \frac{(1 - \mathrm{B}(i)_j)\boldsymbol{b}_j(1 - \boldsymbol{w}_j + \boldsymbol{a}_j) + \mathrm{B}(i)_j \boldsymbol{a}_j(1 - \boldsymbol{w}_j - \boldsymbol{b}_j)}{\boldsymbol{a}_j + \boldsymbol{b}_j} \tag{22}$$

$$= (1 - \boldsymbol{w}_D) \sum_{i=0}^{2^{D-1}-1} \prod_{j=1}^{D-1} \frac{(1 - \mathrm{B}(i)_j)\boldsymbol{b}_j(1 - \boldsymbol{w}_j + \boldsymbol{a}_j) + \mathrm{B}(i)_j \boldsymbol{a}_j(1 - \boldsymbol{w}_j - \boldsymbol{b}_j)}{\boldsymbol{a}_j + \boldsymbol{b}_j}$$

$$\tag{23}$$

$$= \prod_{j=1}^{D} (1 - \boldsymbol{w}_j) = \frac{\partial \phi(\boldsymbol{w}; \boldsymbol{H})}{\partial \boldsymbol{H}_0}, \tag{24}$$

respectively. In a similar way, we can describe the other cases $\boldsymbol{H}_1$ to $\boldsymbol{H}_6$ of the nested trilinear weights since there are two cases for each dimension inside the product. $\square$

**Proposition D.2** (Cubic Bézier curve)**.** *Trilinear interpolation from Definition 4.1 with the weights on the diagonal line $\boldsymbol{t} = (t, t, t)^\intercal$ where $t \in [0, 1]$ is a cubic Bézier curve.*

*Proof.* We rewrite the trilinear interpolation from Definition 4.1 with the weight $\boldsymbol{t}$ as follows:

$$\psi(\boldsymbol{t}; \boldsymbol{H}) = \sum_{i=0}^{2^3-1} \omega(i, \boldsymbol{t}) \cdot \boldsymbol{H}_i \tag{25}$$

$$= \sum_{i=0}^{2^3-1} \prod_{j=1}^{3} \left[ (1 - \mathrm{B}(i)_j)(1 - t) + \mathrm{B}(i)_j t \right] \cdot \boldsymbol{H}_i \tag{26}$$

$$= (1 - t)^3 \boldsymbol{H}_0 + (1 - t)^2 t(\boldsymbol{H}_1 + \boldsymbol{H}_2 + \boldsymbol{H}_4) + (1 - t)t^2(\boldsymbol{H}_3 + \boldsymbol{H}_5 + \boldsymbol{H}_6) + t^3 \boldsymbol{H}_7 \tag{27}$$

$$= (1 - t)^3 \boldsymbol{H}_0 + 3(1 - t)^2 t\left(\frac{\boldsymbol{H}_1 + \boldsymbol{H}_2 + \boldsymbol{H}_4}{3}\right) + 3(1 - t)t^2\left(\frac{\boldsymbol{H}_3 + \boldsymbol{H}_5 + \boldsymbol{H}_6}{3}\right) + t^3 \boldsymbol{H}_7 \tag{28}$$

$$= (1 - t)^3 \boldsymbol{P}_0 + 3(1 - t)^2 t \boldsymbol{P}_1 + 3(1 - t)t^2 \boldsymbol{P}_2 + t^3 \boldsymbol{P}_3 \tag{29}$$

Four points $\boldsymbol{P}_0 := \boldsymbol{H}_0$, $\boldsymbol{P}_1 := \frac{\boldsymbol{H}_1 + \boldsymbol{H}_2 + \boldsymbol{H}_4}{3}$, $\boldsymbol{P}_2 := \frac{\boldsymbol{H}_3 + \boldsymbol{H}_5 + \boldsymbol{H}_6}{3}$, and $\boldsymbol{P}_3 := \boldsymbol{H}_7$ define the cubic Bézier curve. $\square$

As $t$ is changed from zero to one, the curve starts at $P_0$ going toward $P_1$ and turns in the direction of $P_2$ before arriving at $P_3$. Usually, the curve does not pass $P_1$ nor $P_2$; but these give the directional guide for the curve.

**Lemma D.3** (Hypersurfaces in a trilinear space)**.** *In a piecewise trilinear region where $x$ lies, hypersurfaces of the piecewise trilinear networks are defined as follows, following the notation of Definition 4.1:*

$$0 = \tilde{\nu}(\boldsymbol{x}) = \sum_{i=0}^{2^D - 1} \omega(i, \boldsymbol{w}) \cdot \tilde{\nu}(\boldsymbol{x}_i) \tag{30}$$

*where $\boldsymbol{w} \in [0,1]^D$ is the relative position in a grid cell and $\{\boldsymbol{x}_i\}$ and $\{\boldsymbol{H}_i\}$ are the corresponding corner positions and representations, respectively, in a trilinear space.*

*Proof.* In a piecewise trilinear region, we can use the linearity of $\nu$ as follows:

$$\tilde{\nu}(\boldsymbol{x}) = (\nu \circ \tau)(\boldsymbol{x}) = \nu\left( \sum_{i=0}^{2^D - 1} \omega(i, \boldsymbol{w}) \cdot \boldsymbol{H}_i \right) \tag{31}$$

$$= \sum_{i=0}^{2^D - 1} \omega(i, \boldsymbol{w}) \cdot \nu(\boldsymbol{H}_i) \tag{32}$$

$$= \sum_{i=0}^{2^D - 1} \omega(i, \boldsymbol{w}) \cdot \tilde{\nu}(\boldsymbol{x}_i) \tag{33}$$

which concludes the proof. $\qquad\square$

**Lemma D.4** (Curved edge of two hypersurfaces, restated)**.** *In a piecewise trilinear region, let an edge $(\boldsymbol{x}_0, \boldsymbol{x}_7)$ be the intersection of two hypersurfaces $\tilde{\nu}_i(\boldsymbol{x}) = \tilde{\nu}_j(\boldsymbol{x}) = 0$, while $\boldsymbol{x}_0$ and $\boldsymbol{x}_7$ are on the two hypersurfaces. Then, the edge is defined as:*

$$\{\boldsymbol{x} : \tau(\boldsymbol{x}) = (1 - t)\tau(\boldsymbol{x}_0) + t\tau(\boldsymbol{x}_7) \quad where \quad \boldsymbol{x} \in \mathbb{R}^D \quad and \quad t \in [0, 1]\}. \tag{34}$$

*Proof.* By Definition 4.3,

$$\tilde{\nu}_i(\boldsymbol{x}_0) = \tilde{\nu}_i(\boldsymbol{x}_7) \quad \Leftrightarrow \quad \nu_i(\tau(\boldsymbol{x}_0)) = \nu_i(\tau(\boldsymbol{x}_7)), \tag{35}$$

while $\nu_i$ is linear in a piecewise trilinear region. Since it similarly goes for $\nu_j$, the intersection line of two hyperplanes $\nu_i$ and $\nu_j$ is $(1 - t)\tau(\boldsymbol{x}_0) + t\tau(\boldsymbol{x}_7)$ where $t \in \mathbb{R}$. In other words, the edge is on the hypersurface $\nu_i\big((1 - t)\tau(\boldsymbol{x}_0) + t\tau(\boldsymbol{x}_7)\big) = 0$ where $t \in \mathbb{R}$. The edge is defined as follows:

$$\{\boldsymbol{x} : \tau(\boldsymbol{x}) = (1 - t)\tau(\boldsymbol{x}_0) + t\tau(\boldsymbol{x}_7) \quad where \quad \boldsymbol{x} \in \mathbb{R}^D \quad and \quad t \in \mathbb{R}\} \tag{36}$$

which concludes the proof. $\qquad\square$

Notice that when $\tau(\boldsymbol{x})_1 + \tau(\boldsymbol{x})_2 + \tau(\boldsymbol{x})_4 = \tau(\boldsymbol{x})_3 + \tau(\boldsymbol{x})_5 + \tau(\boldsymbol{x})_6 = 0$, at least one connecting edge can be identified. This occurs along the diagonal line $\boldsymbol{x} = t \cdot \boldsymbol{1}$, where $t \in [0, 1]$. Along this line, $\tau(\boldsymbol{x})$ represents a cubic Bézier curve, as shown in Proposition D.2 with the control points of $P_1$ and $P_2$ set to zeros, which is the line.

**Theorem D.5** (Hypersurface and eikonal constraint, restated)**.** *A hypersurface $\tau(\boldsymbol{x}) = 0$ intersects two points $\tau(\boldsymbol{x}_0) = \tau(\boldsymbol{x}_7) = 0$ while $\tau(\boldsymbol{x}_{1\ldots6}) \neq 0$ for the remaining six points. These points form a cube, with $\boldsymbol{x}_0$ and $\boldsymbol{x}_7$ positioned on the diagonal of the cube. The hypersurface satisfies the eikonal constraint $\|\nabla\tau(\boldsymbol{x})\|_2^2 = 1$ for all $\boldsymbol{x} \in [0, 1]^3$. Then, the hypersurface of $\tau(\boldsymbol{x}) = 0$ is a plane.*

*Proof.* By the definitions of $\tau$ in Definition 4.3 and the eikonal constraint,

$$\|\nabla\tau(\boldsymbol{x})\|_2^2 = \left(\frac{\partial\tau(\boldsymbol{x})}{\partial x}\right)^2 + \left(\frac{\partial\tau(\boldsymbol{x})}{\partial y}\right)^2 + \left(\frac{\partial\tau(\boldsymbol{x})}{\partial z}\right)^2 = 1 \tag{37}$$

$$\frac{\partial\|\nabla\tau(\boldsymbol{x})\|_2^2}{\partial x} = 2\left(\frac{\partial\tau(\boldsymbol{x})}{\partial y}\right)\left(\frac{\partial^2\tau(\boldsymbol{x})}{\partial y\partial x}\right) + 2\left(\frac{\partial\tau(\boldsymbol{x})}{\partial z}\right)\left(\frac{\partial^2\tau(\boldsymbol{x})}{\partial z\partial x}\right) = 0 \tag{38}$$

$$\frac{\partial\|\nabla\tau(\boldsymbol{y})\|_2^2}{\partial y} = 2\left(\frac{\partial\tau(\boldsymbol{x})}{\partial x}\right)\left(\frac{\partial^2\tau(\boldsymbol{x})}{\partial x\partial y}\right) + 2\left(\frac{\partial\tau(\boldsymbol{x})}{\partial z}\right)\left(\frac{\partial^2\tau(\boldsymbol{x})}{\partial z\partial y}\right) = 0 \tag{39}$$

$$\frac{\partial\|\nabla\tau(\boldsymbol{z})\|_2^2}{\partial z} = 2\left(\frac{\partial\tau(\boldsymbol{x})}{\partial x}\right)\left(\frac{\partial^2\tau(\boldsymbol{x})}{\partial x\partial z}\right) + 2\left(\frac{\partial\tau(\boldsymbol{x})}{\partial y}\right)\left(\frac{\partial^2\tau(\boldsymbol{x})}{\partial y\partial z}\right) = 0 \tag{40}$$

where

$$\frac{\partial\tau(\boldsymbol{x})}{\partial y} = (1-x)\Big((1-z)\big(\tau(\boldsymbol{x}_2)-\tau(\boldsymbol{x}_0)\big) + z\big(\tau(\boldsymbol{x}_6)-\tau(\boldsymbol{x}_4)\big)\Big)$$
$$+ x\Big((1-z)\big(\tau(\boldsymbol{x}_3)-\tau(\boldsymbol{x}_1)\big) + z\big(\tau(\boldsymbol{x}_7)-\tau(\boldsymbol{x}_5)\big)\Big). \tag{41}$$

However, since $\tau(\boldsymbol{x}_0) = \tau(\boldsymbol{x}_7) = 0$ while $\tau(\boldsymbol{x}_2) \neq 0$ and $\tau(\boldsymbol{x}_5) \neq 0$, there is no solution $\tau(\boldsymbol{x}_{1\ldots6})$ of $\frac{\partial\tau(\boldsymbol{x})}{\partial y} = 0$ for all $\boldsymbol{x}$. Likewise, the same goes for $\frac{\partial\tau(\boldsymbol{x})}{\partial x}$ and $\frac{\partial\tau(\boldsymbol{x})}{\partial z}$. Additionally, the three partial derivatives can be rewritten as follows:

$$bC + cB = aC + cA = aB + bA = 0 \tag{42}$$

where $a \neq 0$, $b \neq 0$, and $c \neq 0$. By substituting $B = -bC/c$ and $A = -aC/c$, we can show that $-abC/c - abC/c = 0 \Leftrightarrow C = 0$, which makes $A = B = C = 0$. Therefore, we rewrite A, B, and C as follows:

$$\frac{\partial^2\tau(\boldsymbol{x})}{\partial y\partial z} = (1-x)\big(-\tau(\boldsymbol{x}_2)-\tau(\boldsymbol{x}_4)+\tau(\boldsymbol{x}_6)\big) + x\big(+\tau(\boldsymbol{x}_1)-\tau(\boldsymbol{x}_3)-\tau(\boldsymbol{x}_5)\big) \tag{43}$$

$$\frac{\partial^2\tau(\boldsymbol{x})}{\partial z\partial x} = (1-y)\big(-\tau(\boldsymbol{x}_1)-\tau(\boldsymbol{x}_4)+\tau(\boldsymbol{x}_5)\big) + y\big(+\tau(\boldsymbol{x}_2)-\tau(\boldsymbol{x}_3)-\tau(\boldsymbol{x}_6)\big) \tag{44}$$

$$\frac{\partial^2\tau(\boldsymbol{x})}{\partial x\partial y} = (1-z)\big(-\tau(\boldsymbol{x}_1)-\tau(\boldsymbol{x}_2)+\tau(\boldsymbol{x}_3)\big) + z\big(+\tau(\boldsymbol{x}_4)-\tau(\boldsymbol{x}_5)-\tau(\boldsymbol{x}_6)\big), \tag{45}$$

respectively. The following conditions would satisfy the eikonal constraint:

$$-\tau(\boldsymbol{x}_2) - \tau(\boldsymbol{x}_4) + \tau(\boldsymbol{x}_6) = 0 \tag{46}$$
$$\tau(\boldsymbol{x}_1) - \tau(\boldsymbol{x}_3) - \tau(\boldsymbol{x}_5) = 0 \tag{47}$$
$$-\tau(\boldsymbol{x}_1) - \tau(\boldsymbol{x}_4) + \tau(\boldsymbol{x}_5) = 0 \tag{48}$$
$$+\tau(\boldsymbol{x}_2) - \tau(\boldsymbol{x}_3) - \tau(\boldsymbol{x}_6) = 0 \tag{49}$$
$$-\tau(\boldsymbol{x}_1) - \tau(\boldsymbol{x}_2) + \tau(\boldsymbol{x}_3) = 0 \tag{50}$$
$$+\tau(\boldsymbol{x}_4) - \tau(\boldsymbol{x}_5) - \tau(\boldsymbol{x}_6) = 0 \tag{51}$$

From Equation (48) and Equation (51),

$$\tau(\boldsymbol{x}_1) = -\tau(\boldsymbol{x}_6) \tag{52}$$

From Equation (47) and Equation (50),

$$\tau(\boldsymbol{x}_2) = -\tau(\boldsymbol{x}_5) \tag{53}$$

From Equation (47) and Equation (48),

$$\tau(\boldsymbol{x}_4) = -\tau(\boldsymbol{x}_3) \tag{54}$$

From Equation (50), Equation (51), and Equation (54),

$$\tau(\boldsymbol{x}_1) + \tau(\boldsymbol{x}_2) + \tau(\boldsymbol{x}_4) = 0 \tag{55}$$
$$\tau(\boldsymbol{x}_3) + \tau(\boldsymbol{x}_5) + \tau(\boldsymbol{x}_6) = 0 \tag{56}$$

Using Equation (52), Equation (53), Equation (54), Equation (55), and Equation (56),

$$
\tau(\boldsymbol{x}) = \big(x(1-y)(1-z) - (1-x)yz\big)\tau(\boldsymbol{x}_1)
$$
$$
+ \big((1-x)y(1-z) - x(1-y)z\big)\tau(\boldsymbol{x}_2)
$$
$$
+ \big((1-x)(1-y)z - xy(1-z)\big)\tau(\boldsymbol{x}_4) \tag{57}
$$
$$
= (x+\alpha)\tau(\boldsymbol{x}_1) + (y+\alpha)\tau(\boldsymbol{x}_2) + (z+\alpha)\tau(\boldsymbol{x}_4) \tag{58}
$$
$$
= x\tau(\boldsymbol{x}_1) + y\tau(\boldsymbol{x}_2) + z\tau(\boldsymbol{x}_4) + \alpha\big(\tau(\boldsymbol{x}_1) + \tau(\boldsymbol{x}_2) + \tau(\boldsymbol{x}_4)\big) \tag{59}
$$
$$
= x\tau(\boldsymbol{x}_1) + y\tau(\boldsymbol{x}_2) + z\tau(\boldsymbol{x}_4) = 0 \tag{60}
$$

where $\alpha = 2xyz - xy - yz - zx$.

This is an equation of a plane where its normal vector is $\big(\tau(\boldsymbol{x}_1), \tau(\boldsymbol{x}_2), \tau(\boldsymbol{x}_4)\big)$. Again, to satisfy the eikonal constraint,

$$
\tau(\boldsymbol{x}_1)^2 + \tau(\boldsymbol{x}_2)^2 + \tau(\boldsymbol{x}_4)^2 = 1 \tag{61}
$$
$$
\tau(\boldsymbol{x}_3)^2 + \tau(\boldsymbol{x}_5)^2 + \tau(\boldsymbol{x}_6)^2 = 1, \tag{62}
$$

which concludes the proof. $\qquad\square$

**Corollary D.6** (Affine-transformed hypersurface and eikonal constraint, restated). *Let $\nu_0(\boldsymbol{x})$ and $\nu_1(\boldsymbol{x})$ be two affine transformations defined by $\boldsymbol{W}_i^\mathsf{T}\boldsymbol{x} + \boldsymbol{b}_i$, $i \in \{0,1\}$. A hypersurface $\nu_0\big(\tau(\boldsymbol{x})\big) = 0$ passing two points $\nu_0\big(\tau(\boldsymbol{x}_0)\big) = \nu_0\big(\tau(\boldsymbol{x}_7)\big) = 0$ while $\nu_0\big(\tau(\boldsymbol{x}_{1\dots6})\big) \neq 0$ satisfies the eikonal constraint $\|\nabla\nu_1\big(\tau(\boldsymbol{x})\big)\|_2^2 = 1$ for all $\boldsymbol{x} \in [0,1]^3$. Then, the hypersurface is a plane.*

*Proof.* Similarly to Theorem D.5, we investigate the Jacobian of the eikonal constraint to be zero as follows:

$$
\|\nabla\nu_1\big(\tau(\boldsymbol{x})\big)\|_2 = \Big(\frac{\partial\nu_1}{\partial\tau}\frac{\partial\tau(\boldsymbol{x})}{\partial x}\Big)^2 + \Big(\frac{\partial\nu_1}{\partial\tau}\frac{\partial\tau(\boldsymbol{x})}{\partial y}\Big)^2 + \Big(\frac{\partial\nu_1}{\partial\tau}\frac{\partial\tau(\boldsymbol{x})}{\partial z}\Big)^2 = 1 \tag{63}
$$
$$
\frac{\partial\|\nabla\nu_1\big(\tau(\boldsymbol{x})\big)\|_2}{\partial x} = 2\Big(\frac{\partial\nu_1}{\partial\tau}\frac{\partial\tau(\boldsymbol{x})}{\partial y}\Big)\Big(\frac{\partial\nu_1}{\partial\tau}\frac{\partial^2\tau(\boldsymbol{x})}{\partial y\partial x}\Big) + 2\Big(\frac{\partial\nu_1}{\partial\tau}\frac{\partial\tau(\boldsymbol{x})}{\partial z}\Big)\Big(\frac{\partial\nu_1}{\partial\tau}\frac{\partial^2\tau(\boldsymbol{x})}{\partial z\partial x}\Big) = 0. \tag{64}
$$

Since $\partial\nu_1/\partial\tau$ is a constant, using Lemma D.3 and replacing $\nu_1$ and $\tau$ with $\nu_1 \circ \nu_0^{-1}$ and $\nu_0 \circ \tau$, respectively, give us the same constaints for $\nu_0\big(\tau(\boldsymbol{x}_{1\dots6})\big)$ regardless of $\nu_1$. $\qquad\square$

**Theorem D.7** (Intersection of two hypersurfaces and a diagonal plane, restated). *Let $\tilde{\nu}_0(\boldsymbol{x}) = 0$ and $\tilde{\nu}_1(\boldsymbol{x}) = 0$ be two hypersurfaces passing two points $\boldsymbol{x}_0$ and $\boldsymbol{x}_7$ such that $\tilde{\nu}_0(\boldsymbol{x}_0) = \tilde{\nu}_1(\boldsymbol{x}_0) = 0$ and $\tilde{\nu}_0(\boldsymbol{x}_7) = \tilde{\nu}_1(\boldsymbol{x}_7) = 0$, $\boldsymbol{P}_i := \tilde{\nu}_0(\boldsymbol{x}_i)$ and $\boldsymbol{Q}_i := \tilde{\nu}_1(\boldsymbol{x}_i)$,*

$$
\boldsymbol{P}_\alpha = \big[\boldsymbol{P}_0; \ \boldsymbol{P}_1; \ \boldsymbol{P}_4; \ \boldsymbol{P}_5\big], \qquad \boldsymbol{P}_\beta = \big[\boldsymbol{P}_2; \ \boldsymbol{P}_3; \ \boldsymbol{P}_6; \ \boldsymbol{P}_7\big], \tag{65}
$$

*and*

$$
\boldsymbol{X} = \begin{bmatrix} (1-x)^2 \\ x(1-x) \\ (1-x)x \\ x^2 \end{bmatrix}. \tag{66}
$$

*Then, $x \in [0,1]$ of the intersection point of the two hypersurfaces and a diagonal plane of $x = z$ is the solution of the following quartic equation:*

$$
\boldsymbol{X}^\mathsf{T}\big(\boldsymbol{P}_\alpha\boldsymbol{Q}_\beta^\mathsf{T} - \boldsymbol{P}_\beta\boldsymbol{Q}_\alpha^\mathsf{T}\big)\boldsymbol{X} = 0 \tag{67}
$$

*while*

$$
y = \frac{\boldsymbol{X}^\mathsf{T}\boldsymbol{P}_\alpha}{\boldsymbol{X}^\mathsf{T}(\boldsymbol{P}_\alpha - \boldsymbol{P}_\beta)} \qquad (\boldsymbol{P}_\alpha \neq \boldsymbol{P}_\beta). \tag{68}
$$

*Proof.* We rearrange $\tilde{\nu}_0(\boldsymbol{x})$ using $x = z$ as follows:

$$
\tilde{\nu}_0(\boldsymbol{x}) = (1-y) \cdot \boldsymbol{X}^\mathsf{T}\big[\boldsymbol{P}_0; \ \boldsymbol{P}_1; \ \boldsymbol{P}_4; \ \boldsymbol{P}_5\big] + y \cdot \boldsymbol{X}^\mathsf{T}\big[\boldsymbol{P}_2; \ \boldsymbol{P}_3; \ \boldsymbol{P}_6; \ \boldsymbol{P}_7\big] \tag{69}
$$
$$
= (1-y)\boldsymbol{X}^\mathsf{T}\boldsymbol{P}_\alpha + y\boldsymbol{X}^\mathsf{T}\boldsymbol{P}_\beta = 0 \tag{70}
$$
$$
\Leftrightarrow y = \frac{\boldsymbol{X}^\mathsf{T}\boldsymbol{P}_\alpha}{\boldsymbol{X}^\mathsf{T}(\boldsymbol{P}_\alpha - \boldsymbol{P}_\beta)} \qquad (\boldsymbol{P}_\alpha \neq \boldsymbol{P}_\beta). \tag{71}
$$

In a similar way, we rewrite $\tilde{\nu}_1(\boldsymbol{x})$ as follows:

$$\boldsymbol{X}^\intercal(\boldsymbol{P}_\alpha - \boldsymbol{P}_\beta)\tilde{\nu}_1(\boldsymbol{x}) = \boldsymbol{X}^\intercal \boldsymbol{P}_\alpha \cdot \boldsymbol{X}^\intercal \boldsymbol{Q}_\beta - \boldsymbol{X}^\intercal \boldsymbol{P}_\beta \cdot \boldsymbol{X}^\intercal \boldsymbol{Q}_\alpha \tag{72}$$

$$= \boldsymbol{X}^\intercal \boldsymbol{P}_\alpha \boldsymbol{Q}_\beta^\intercal \boldsymbol{X} - \boldsymbol{X}^\intercal \boldsymbol{P}_\beta \boldsymbol{Q}_\alpha^\intercal \boldsymbol{X} \tag{73}$$

$$= \boldsymbol{X}^\intercal \big(\boldsymbol{P}_\alpha \boldsymbol{Q}_\beta^\intercal - \boldsymbol{P}_\beta \boldsymbol{Q}_\alpha^\intercal\big) \boldsymbol{X}. \tag{74}$$

Notice that this is a *quartic* equation of $x$. To find the coefficients of the general form of quadratic equation, we consider a mapping $\boldsymbol{C}$ from the interpolation coefficients $\tau_i$ to polynomial coefficients $c_i$:

$$\begin{bmatrix} c_0 \\ c_1 \\ c_2 \end{bmatrix} = \boldsymbol{C} \begin{bmatrix} \tau_0 \\ \tau_1 \\ \tau_2 \\ \tau_3 \end{bmatrix} = \begin{bmatrix} 1 & 0 & 0 & 0 \\ -2 & 1 & 1 & 0 \\ 1 & -1 & -1 & 1 \end{bmatrix} \begin{bmatrix} \tau_0 \\ \tau_1 \\ \tau_2 \\ \tau_3 \end{bmatrix} \tag{75}$$

where $\tau_0(1-x)^2 + \tau_1 x(1-x) + \tau_2(1-x)x + \tau_3 x^2 = c_0 + c_1 x + c_2 x^2$. Then, the coefficients of the quartic equation of Equation (74) are as follows:

$$\begin{bmatrix} c_{00} & c_{01} & c_{02} \\ c_{10} & c_{11} & c_{12} \\ c_{20} & c_{21} & c_{22} \end{bmatrix} = \boldsymbol{C}\big(\boldsymbol{P}_\alpha \boldsymbol{Q}_\beta^\intercal - \boldsymbol{P}_\beta \boldsymbol{Q}_\alpha^\intercal\big)\boldsymbol{C}^\intercal \tag{76}$$

where the anti-diagonal sum of the result is the quartic coefficients, *i.e.*,

$$c_{00} + (c_{10} + c_{01})x + (c_{20} + c_{11} + c_{02})x^2 + (c_{21} + c_{12})x^3 + c_{22}x^4. \tag{77}$$

The solutions of the quartic equation such that $x \in [0, 1]$ are the valid intersection points, and we choose one of them $x$ if there are more than one solution. Note that we can compute a batch of quartic equations using the parallel computing of the eigenvalue decomposition with Lemma D.8.

We determine $y$ by Equation (71) using $x$, and $z = x$. □

We reproduce the method to find polynomial roots from companion matrix eigenvalues [29].

**Lemma D.8** (Polynomial roots from companion matrix eigenvalues, reproduced). *The companion matrix of the monic polynomial* [7]

$$p(x) = c_0 + c_1 x + \cdots + c_{n-1} x^{n-1} + x^n \tag{78}$$

*is defined as:*

$$C(p) = \begin{bmatrix} 0 & 0 & \cdots & 0 & -c_0 \\ 1 & 0 & \cdots & 0 & -c_1 \\ 0 & 1 & \cdots & 0 & -c_2 \\ \vdots & \vdots & \ddots & \vdots & \vdots \\ 0 & 0 & \cdots & 1 & -c_{n-1} \end{bmatrix}. \tag{79}$$

*The roots of the characteristic polynomial $p(x)$ are the eigenvalues of $C(p)$.*

---

[7]The nonzero coefficient of the highest degree is one. If you are dealing with the equation, you can make it by dividing the nonzero coefficient of the highest degree for both sides.

# E   Complexity analysis

We provide a detailed exposition of the complexity analysis in Section 3.2 to ensure the comprehensiveness of our work. Notice that our complexity analysis closely aligns with the approach outlined in Berzins [4], as in their Appendix A.

Algorithm 2 operates on the vertices and edges, $\mathcal{V}$ and $\mathcal{E}$, respectively. Therefore, our initial focus is on the complexity associated with these inputs. Let $|\mathcal{V}|$, $|\mathcal{E}|$, and $|\hat{\mathcal{E}}|$ ($\hat{\mathcal{E}} \subset \mathcal{E}$) be the number of vertices, edges, and splitting edges consisting of polygons (Section 3.2), at the iteration of $i$ and $j$.

1. During the initialization step (refer to Section 5), we determine $|\mathcal{V}|$ and $|\mathcal{E}|$ to construct a grid in $\mathbb{R}^D$. In the case of having $M$ HashGrid marks, the quantities are defined as $M^D$ and $D(M-1)M^{D-1}$, respectively. In case of the large $M$, we consider vertex and edge pruning for efficient computations inspired by a pruning strategy proposed in Berzins [4]. If, for every future neuron, denoted as $\phi(i,j) > \phi(i^*, j^*)$, the two vertices of an edge exhibit identical signs, the edge can be safely pruned as it will not undergo splitting. We make the assumption of that $M^D < |\mathcal{V}|$ to avoid an excessively dense grid, with a complexity of $\mathcal{O}(|\mathcal{V}|)$.

2. Evaluate $\tilde{\nu}(\mathcal{V}; i, j)$ with a complexity of $\mathcal{O}(|\mathcal{V}|)$, assuming negligible inference costs as constants. The intermediate results from the inference can be used for the sign-vectors (Definition 3.4).

3. Identify splitting edges $\hat{\mathcal{E}}$ by comparing two signs per edge, with a complexity of $\mathcal{O}(|\mathcal{E}|)$.

4. Obtain $\boldsymbol{P}$ and $\boldsymbol{Q}$ for Theorem 4.7 with a complexity of $\mathcal{O}(|\hat{\mathcal{E}}|)$ in a similar way.

5. Inference for Theorem 4.7 involves matrix-vector multiplications and the eigenvalue decomposition of a companion matrix for each edge, with a complexity of $\mathcal{O}(|\hat{\mathcal{E}}|)$. Considering the maximum size of 4 for a quartic equation, this operation is regarded as linear.

6. Find intersecting polygons by pairing two sign-vectors with two common zeros in a shared region of the new vertices resulting from splitting, with a complexity of $\mathcal{O}(|\hat{\mathcal{E}}|)$.

In the study by Berzins [4], a linear relationship between $|\mathcal{V}|$ and $|\mathcal{E}|$ for a mesh is empirically observed (see Figure 8a in Berzins [4]). It is worth to note that the number of splitting edges can be effectively upper-bounded as $|\hat{\mathcal{E}}| < |\mathcal{E}|D/\phi(i,j)$ using Theorem 5 in [20]. Consequently, all the steps outlined in Algorithm 2 demonstrate a linear dependency on the number of vertices, denoted as $\mathcal{O}(|\mathcal{V}|)$.

To obtain faces, we sort the vertices within each region based on the Jacobian with respect to the mean of the vertices (see Algorithm 3). Considering triangularization, where each face has three vertices, the complexity remains $\mathcal{O}(|\mathcal{V}|)$, assuming the negligible cost of calculating the Jacobian, especially when performed in parallel.

# F Supplementary experiments

## F.1 Complete results

Table 2 and Table 3 are extended from the Stanford 3D Scanning dataset in Table 1. Table 4 extended from the Stanford bunny of Table 1, including standard deviations, and pseudo-ground truth results, obtained through marching cubes with 256 samples. Notice that we used the more excessive 512 samples for the *Medium* and *Large* models in Figure 4. We found that 256 samples were not enough for those. Table 5 shows the angular distance results. Tables 2 to 5 use the *Small* model, while Table 8 uses the *Large* model (*ref.* Section 6). We conducted all experiments with an NVIDIA V100 32GB.

Table 2: Chamfer distance (CD) and chamfer efficiency (CE) for the *Part A* of the Stanford 3D Scanning repository [31] using the *Small* model as the default setting (*ref.* Section 6). The chamfer distance ($\times$1e-6) is evaluated using the marching cubes (MC) with $256^3$ grid samples as the reference ground truth for all measurements. The chamfer efficiency is defined as $100/(\sqrt{|\mathcal{V}| \times \text{CD}})$, providing a concise representation of the trade-off between the number of vertices and the chamfer distance. The results are averaged over three random seeds.

| MC # | Bunny | | | Dragon | | | Happy Buddha | | |
|---|---|---|---|---|---|---|---|---|---|
| | $|\mathcal{V}|\downarrow$ | CD $\downarrow$ | CE $\uparrow$ | $|\mathcal{V}|\downarrow$ | CD $\downarrow$ | CE $\uparrow$ | $|\mathcal{V}|\downarrow$ | CD $\downarrow$ | CE $\uparrow$ |
| 32 | 1283 | 4106 | 19.0 | 1416 | 6330 | 11.2 | 1032 | 6951 | 13.9 |
| 64 | 5367 | 1371 | 13.6 | 6090 | 1936 | 8.5 | 4362 | 2465 | 9.3 |
| 128 | 21825 | 393 | 11.6 | 25236 | 542 | 7.3 | 18219 | 722 | 7.6 |
| 192 | 49569 | 141 | 14.3 | 57341 | 203 | 8.6 | 41351 | 276 | 8.8 |
| Ours | 4341 | 900 | **25.6** | 5104 | 1243 | **15.8** | 3710 | 1738 | **15.5** |

Table 3: Chamfer distance (CD) and chamfer efficiency (CE) for the *Part B* of the Stanford 3D Scanning dataset [31].

| MC # | Armadillo | | | Drill Bit | | | Lucy | | |
|---|---|---|---|---|---|---|---|---|---|
| | $|\mathcal{V}|\downarrow$ | CD $\downarrow$ | CE $\uparrow$ | $|\mathcal{V}|\downarrow$ | CD $\downarrow$ | CE $\uparrow$ | $|\mathcal{V}|\downarrow$ | CD $\downarrow$ | CE $\uparrow$ |
| 32 | 1438 | 6212 | 11.2 | 144 | 91959 | 7.5 | 594 | 6753 | **24.9** |
| 64 | 6151 | 1960 | 8.3 | 873 | 5949 | 19.2 | 2701 | 2310 | 16.0 |
| 128 | 25164 | 605 | 6.6 | 3718 | 1542 | 17.4 | 11255 | 830 | 10.7 |
| 192 | 57306 | 231 | 7.5 | 8382 | 782 | 15.3 | 25589 | 335 | 11.7 |
| Ours | 4783 | 1499 | **13.9** | 694 | 3013 | **47.8** | 2406 | 1763 | 23.6 |

Table 4: Chamfer distance (CD) and chamfer efficiency (CE) for the Stanford bunny are evaluated using the marching cubes with $256^3$ grid samples ($\star$) as the reference ground truth. The standard deviations are provided after the $\pm$ notation, based on results obtained from three random seeds, which are used in training. Similar outcomes are observed for the remaining cases.

| Method | Sample | Vertices $\downarrow$ | CD ($\times$1e-6) $\downarrow$ | CE $\uparrow$ | Time $\downarrow$ |
|---|---|---|---|---|---|
| | 32 | $1283 \pm 89$ | $4106 \pm 202$ | 19.0 | $0.00 \pm 0.00$ |
| | 48 | $2967 \pm 177$ | $2178 \pm 94$ | 15.5 | $0.01 \pm 0.06$ |
| | 56 | $4057 \pm 245$ | $1539 \pm 100$ | 16.0 | $0.01 \pm 0.00$ |
| Marching Cubes | 64 | $5367 \pm 342$ | $1371 \pm 93$ | 13.6 | $0.02 \pm 0.00$ |
| | 128 | $21825 \pm 1336$ | $393 \pm 19$ | 11.6 | $0.10 \pm 0.00$ |
| | 192 | $49569 \pm 3043$ | $141 \pm 4$ | 14.3 | $0.44 \pm 0.10$ |
| | 224 | $67601 \pm 4146$ | $114 \pm 6$ | 12.9 | $0.60 \pm 0.03$ |
| Marching Cubes$^\star$ | 256 | $88492 \pm 5404$ | - | - | $0.88 \pm 0.05$ |
| Ours | - | $4341 \pm 253$ | $900 \pm 171$ | **25.6** | $0.97 \pm 0.21$ |

Table 5: Angular distance (AD) for the Stanford bunny. Similarly to the chamfer distance, the angular distance (°) is also evaluated using the marching cubes with $256^3$ grid samples ($\star$) as the reference ground truth for all measurements. The standard deviations are provided after the $\pm$ notation, based on results obtained from three random seeds, which is used in training.

| Method | Sample | Vertices ↓ | AD (°) ↓ |
|---|---|---|---|
| | 32 | 1283 ± 89 | 6.97 ± 0.45 |
| | 48 | 2967 ± 177 | 5.07 ± 0.15 |
| | 56 | 4057 ± 245 | 4.23 ± 0.25 |
| Marching Cubes | 64 | 5367 ± 342 | 4.10 ± 0.17 |
| | 128 | 21825 ± 1336 | 2.47 ± 0.12 |
| | 192 | 49569 ± 3043 | 1.63 ± 0.06 |
| | 224 | 67601 ± 4146 | 1.53 ± 0.06 |
| Marching Cubes$^\star$ | 256 | 88492 ± 253 | - |
| Ours | - | 4490 ± 91 | 3.57 ± 0.38 |

Table 6: Chamfer distance (CD) and chamfer efficiency (CE) for the *Part A* of the Stanford 3D Scanning repository [31] using the *Large* model having 4x-resolution of HashGrid compared to the default setting (*ref.* Section 6). The chamfer distance (×1e-6) is evaluated using the marching cubes (MC) with $512^3$ grid samples as the reference ground truth for all measurements. The results are averaged over three random seeds.

| | Bunny | | | Dragon | | | Happy Buddha | | |
|---|---|---|---|---|---|---|---|---|---|
| MC # | $|\mathcal{V}|$ ↓ | CD ↓ | CE ↑ | $|\mathcal{V}|$ ↓ | CD ↓ | CE ↑ | $|\mathcal{V}|$ ↓ | CD ↓ | CE ↑ |
| 128 | 38395 | 632 | 4.12 | 25986 | 759 | 5.07 | 18621 | 1110 | 4.84 |
| 192 | 87245 | 299 | 3.83 | 58855 | 335 | 5.08 | 42793 | 533 | 4.38 |
| 224 | 119000 | 221 | 3.80 | 80470 | 268 | 4.64 | 58357 | 395 | 4.34 |
| 256 | 155484 | 182 | 3.53 | 105235 | 226 | 4.21 | 76183 | 322 | 4.07 |
| Ours | 135691 | **151** | **4.87** | 91147 | **157** | **6.99** | 67562 | **229** | **6.46** |

Table 7: Chamfer distance (CD) and chamfer efficiency (CE) for the *Part B* of the Stanford 3D Scanning dataset [31] using the *Large* model as the default setting (*ref.* Section 6).

| | Armadillo | | | Drill Bit | | | Lucy | | |
|---|---|---|---|---|---|---|---|---|---|
| MC # | $|\mathcal{V}|$ ↓ | CD ↓ | CE ↑ | $|\mathcal{V}|$ ↓ | CD ↓ | CE ↑ | $|\mathcal{V}|$ ↓ | CD ↓ | CE ↑ |
| 128 | 25309 | 887 | 4.45 | 3590 | 1342 | 20.8 | 12224 | 1304 | 6.28 |
| 192 | 57751 | 399 | 4.34 | 8092 | 625 | 19.8 | 27863 | 687 | 5.23 |
| 224 | 78703 | 315 | 4.04 | 11000 | 588 | 15.5 | 38337 | 524 | 4.97 |
| 256 | 103085 | 267 | 3.63 | 14339 | 496 | 14.1 | 49838 | 435 | 4.61 |
| Ours | 90567 | **186** | **5.94** | 10275 | **342** | **28.5** | 43631 | **358** | **6.40** |

Table 8: Chamfer distance (CD) and chamfer efficiency (CE) for the Stanford dragon using the *Large* model having 4x-resolution of HashGrid compared to the default setting (*ref.* Section 6) are evaluated using the marching cubes with $512^3$ grid samples ($\star$) as the reference ground truth.

| Method | Sample | Vertices ↓ | CD (×1e-6) ↓ | CE ↑ | Time ↓ |
|---|---|---|---|---|---|
| | 128 | 25986 ± 121 | 759 ± 64 | 5.07 | 0.29 ± 0.01 |
| Marching Cubes | 192 | 58855 ± 181 | 335 ± 21 | 5.08 | 0.60 ± 0.06 |
| | 256 | 105235 ± 452 | 226 ± 17 | 4.21 | 1.16 ± 0.04 |
| Marching Cubes$^\star$ | 512 | 424311 ± 1262 | - | - | 9.19 ± 0.39 |
| Ours | - | 91147 ± 203 | **157** ± 18 | **6.99** | 14.9 ± 1.66 |

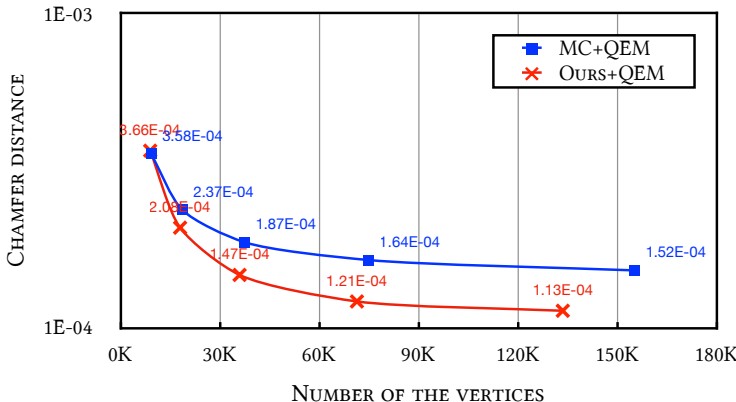

Figure 8: Chamfer distance for the Bunny with the *Large* model applying QEM (a mesh simplification). Both benefit from QEM while our method efficiently retains better chamfer distances.

## F.2  Mesh simplification using QEM

In Figure 8, Quadric Error Metrics (QEM) [35] is a popular mesh simplification algorithm, it could potentially improve both MC and our method for efficiency. Overall, as shown, the mesh simplification favors our method. Notice that QEM entails a higher computational cost, at least $O(n \log n)$ where $n$ is the number of vertices. Additionally, these methods can cause shape deformation or holes, sensitive to hyperparameters. Therefore, QEM may be applied with caution for further mesh simplification, acknowledging the potential for unpredictable side effects. We used MeshLab's Quadric Edge Collapse Decimation with the default settings to halve the number of vertices for each step.

## F.3  Comparison with alternate approaches

Marching Cubes (MC) remains the *de facto* sampling-based method. In contrast, our method is a white-box method grounded in the theoretical understanding of trilinear interpolation and polyhedral complex derivation. Here is our rationale for this evaluation:

1. Analytical approaches for CPWA functions are limited by their inability to effectively manage spectral bias and their exponential computational cost, which struggles with exponentially growing linear regions (infeasible to run our benchmarks.) For example, the time complexity of Analytic Marching [2] is $\mathcal{O}((n/2)^{2(L-1)}|\mathcal{V}_\mathcal{P}|n^4 L^2)$, which grows exponentially with the number of layers $L$ and the width $n$ of the networks, where $|\mathcal{V}_\mathcal{P}|$ represents the number of vertices per face (Section 4.1 from Lei and Jia [2]). In contrast, our method has a linear time complexity with respect to the number of vertices, as discussed in Appendix E, allowing it to avoid visiting every linear region exponentially growing.

2. We also exclude optimization-based methods as they typically rely on MC or its variants initialization in their pipelines (our method can also be integrated into them), such as MeshSDF [36], Deep Marching Tetrahedra [10], and FlexiCubes [37]. Additionally, these methods result in prolonged computational costs.

We provided both rigorous quantitative Figure 3 and qualitative Figure 6 analyses on Marching Tetrahedra (MT) [9, 10] and Neural Dual Contour (NDC) [32], which serve as alternative representative approaches. MT is a popular variant of MC that utilizes six tetrahedra within each grid to identify intermediate vertices. NDC is a data-driven approach (a pretrained model) that uses Dual Contour. In short, MT efficiently requires SDF values but is less efficient regarding the number of vertices extracted. (MT is better than MC in the same resolution, producing more vertices, though.) NDC faces a generalization issue for a zero-shot setting, producing unsmooth surfaces. We used the code from Deep Marching Tetrahedra [10] for MT and the official code of NDC. And PyMCubes for our modernized MC library.

### F.4    Error of the learned SDF

In Equation (11) and Figure 5, we confirmed our hypothesis that the eikonal loss induces piecewise linearity in trilinear networks. To further support this, we also present the error analysis of the learned SDF. Although the Chamfer distance virtually evaluates the SDF values based on sampling, concerns may arise regarding noise in the pseudo-ground truth. For this reason, one might wish to verify whether the predicted SDFs converge toward near-zero.

We randomly sampled 100K points on the surface of pretrained networks (the Stanford bunny *Large* model) and calculated the average of squares for predicted SDF (model's outputs) on the surface. As we expected, the predicted SDFs on the reconstructed surface converge to zero as the weight of the eikonal loss increases, and the number is close to zero $\approx$ 3e-8 as shown in Table 9. Note that, for a *Large* model, fine-grained trilinear regions from the dense grid marks (from the more dense multi-resolution grids) allow relatively small errors of $97 \times$ 1e-9 without the eikonal loss, although the eikonal loss of 1e-2 further decreases to $31 \times$ 1e-9. Note that the stronger eikonal loss of 1e-1 proved ineffective due to over-regularization.

Table 9: Error of the learned SDF. *Weight* stands for the weight of the eikonal loss and *SE* stands for the squared error of predicted SDFs.

| Method | Weight | SE (1e-9) $\downarrow$ |
|---|---|---|
| MC64 | 1e-2 | 2308 |
| MC256 | 1e-2 | 66 |
| | 0 | 97 |
| | 1e-4 | 91 |
| Ours | 1e-3 | 68 |
| | 1e-2 | **31** |

### F.5    Effect of model sizes

In Table 10, we explore the ablation study to determine whether model size impacts model fitting, which in turn affects the GT mesh (we used MC256 for the pseudo-ground truth of the *Small* models). With a smaller model size, the learned mesh becomes simpler due to underfitting. As our evaluation focuses on how accurately the extraction methods capture the underlying zero level-set decision boundary of the networks, the change in CD remains small.

Table 10: Ablation study varying the number of *layers* and the *width* of networks for the Stanford bunny *Small* models. And, *Note* indicates the architecture of the decoding networks.

| Layers | Width | Vertices | CD (1e-6) $\downarrow$ | Time (sec) $\downarrow$ | Note |
|---|---|---|---|---|---|
| 2 | 4 | 4565 | 738 | 0.11 | |
| 2 | 8 | 4863 | 768 | 0.13 | |
| 2 | 16 | 4566 | 734 | 0.12 | |
| 2 | 32 | 4577 | 754 | 0.22 | |
| 2 | 64 | 4611 | 739 | 0.50 | *InstantNGP* [12] |
| 3 | 4 | 4547 | 666 | 0.12 | |
| 3 | 8 | 4661 | 750 | 0.19 | |
| 3 | 16 | 4628 | 724 | 0.65 | *Ours* |
| 3 | 32 | 4567 | 754 | 0.57 | |
| 3 | 64 | 5588 | 759 | 2.86 | |

Notice that the model with the number of layers of 3 and the width of 64 has the longest runtime of 2.80, primarily because the number of intermediate vertices exceeds 80K. The time complexity is more sensitive to the number of vertices than to the number of layers or the width, which are only indirect factors. Note that the model capacity of the decoding networks is comparable to InstantNGP [12], where the representational efficiency is driven by the HashGrid.

# G  Visualizations

For Figure 6 in the main paper (closer looks at bunny's nose), we want to leave a note about the small facets along the left-side of the nose line in the highlighted region. We assume these narrow facets are embedded in the facets of MC256 (pseudo-ground truth), while MC64 fails to capture the left and right edge of the nose and represents this with more facets cutting these two sides of edges – not aligned with nose lines and rather assign more facets in here.

In Figure 9, we explain the rationale behind adopting the diagonal plane assumption in Theorem 4.7 and Section 4.2. We illustrated 64 edge scenarios within a trilinear region, omitting isomorphic cases.

In Figure 10, we showcase the analytical generation of a normal map from piecewise trilinear networks, utilizing HashGrid [12] and ReLU neural networks. Our approach dynamically assigns vertices based on the learned decision boundary. Notably, the configuration of hash grids influences vertex selection, and each grid can be subdivided into multiple polyhedra for accurate representation of smooth curves.

In Figure 11, we present comparative visualizations of normal maps for marching cubes, demonstrating variations in the number of samplings alongside our method, with the respective vertex counts indicated in parentheses. This analysis highlights the trade-off between detail preservation and computational efficiency, showcasing the optimality of our analytically extracted mesh from decision boundary apex points.

In Figure 12, we compare marching cubes (MC) with 256 and 64 samplings to our method. MC 64 shows limitations in representing pointy areas, while our approach excels with analytical extraction from decision boundary apex points. Adaptive sampling leads to some vertices being in close proximity (10.1% within 1e-3), prompting potential mesh optimization for improved efficiency.

In Figure 13, the gallery of the six Stanford 3D Scanning meshes from the *Large* models is shown.

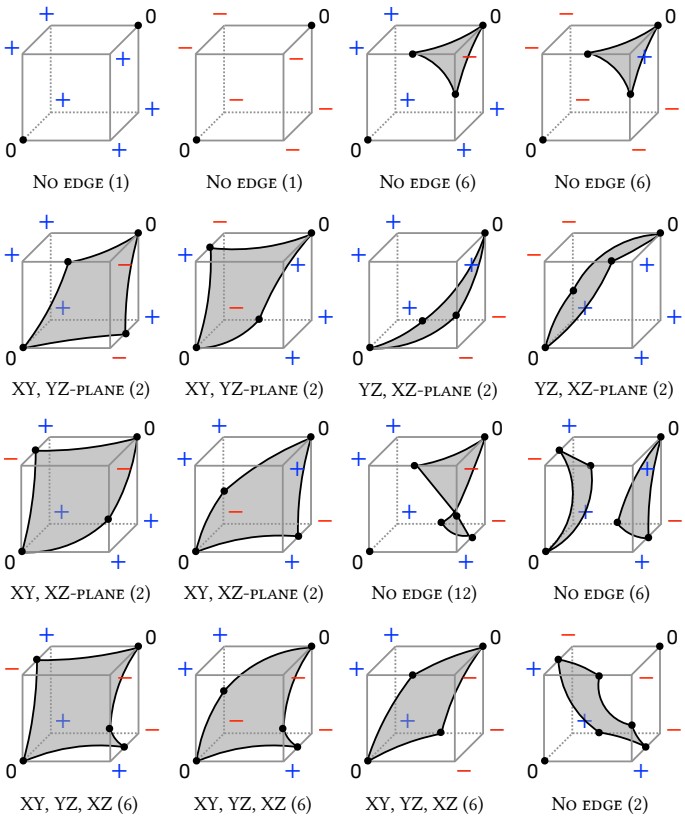

Figure 9: We depict the 64 scenarios of the edge within a trilinear region, excluding isomorph cases specifying the count within parentheses. *No edge* indicates where two diagonal zero corners are not connected. The labels *XY*, *YZ*, and *XZ* refer to $x = y$, $y = z$, and $z = x$ diagonal planes, respectively, form the shared edge, connecting the two diagonal zero corners, with the hypersurface in the schematic figure. In nearly all cases, a diagonal plane intersects with the hypersurface, except, at most, four cases, depending on the curvature determined by corner values.

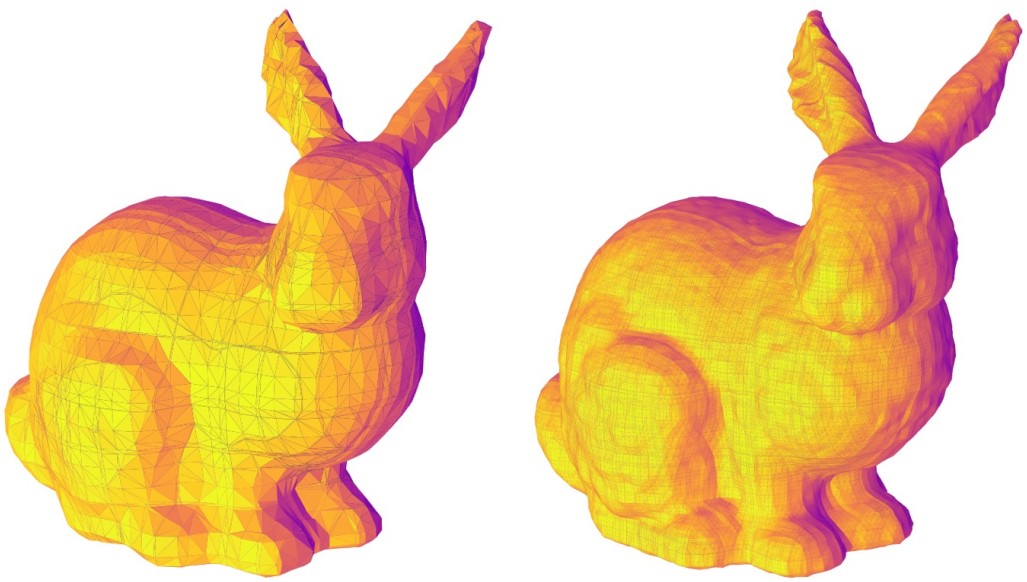

Figure 10: The normal map is analytically generated from piecewise trilinear networks, comprising both HashGrid [12] and ReLU neural networks, that have been trained to learn an SDF with the eikonal loss for the Stanford bunny [31]. Our approach dynamically assigns vertices based on the learned decision boundary. One notable observation is that the hash grids influence the selection of vertices. Additionally, each grid can be subdivided into multiple polyhedra to accurately represent smooth curves. The *Small* (left) and *Large* (right) models (*ref.* Section 6) are used to learn the SDF.

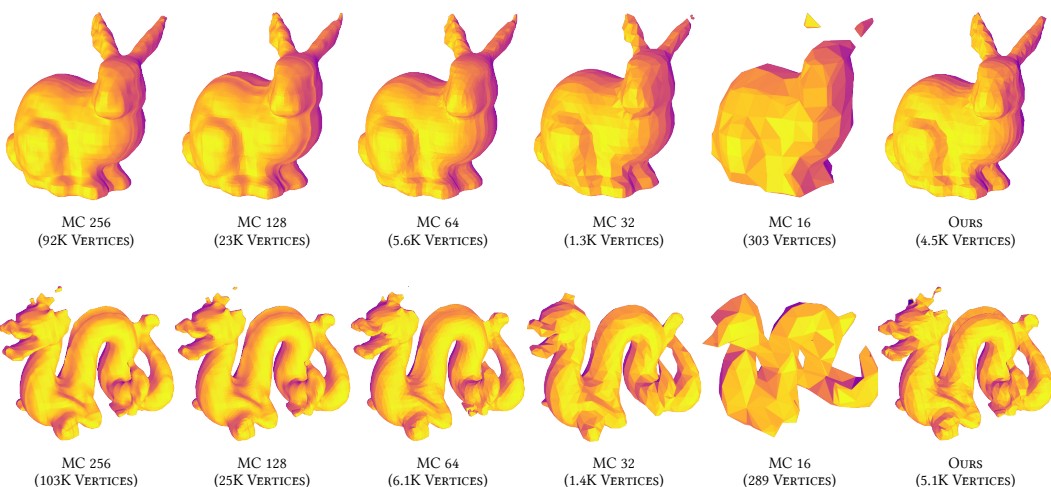

| MC 256 | MC 128 | MC 64 | MC 32 | MC 16 | OURS |
| (92K VERTICES) | (23K VERTICES) | (5.6K VERTICES) | (1.3K VERTICES) | (303 VERTICES) | (4.5K VERTICES) |

| MC 256 | MC 128 | MC 64 | MC 32 | MC 16 | OURS |
| (103K VERTICES) | (25K VERTICES) | (6.1K VERTICES) | (1.4K VERTICES) | (289 VERTICES) | (5.1K VERTICES) |

Figure 11: Visualizing normal maps for marching cubes varying the number of samplings, and ours, with the number of vertices denoted in parentheses. Low sampling has a risk of omitting sharp object details, while dense sampling results in redundant vertices and faces. As our mesh is analytically extracted from the decision boundary apex points of networks, it represents an optimal choice in balancing detail preservation and efficiency.

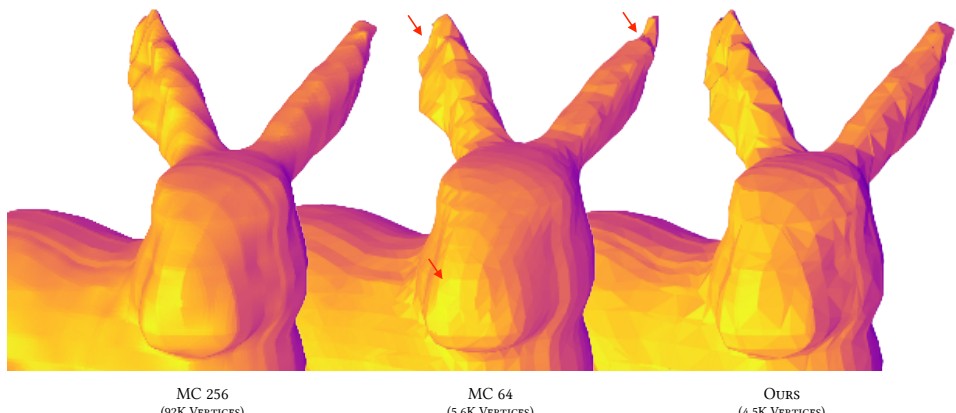

| MC 256 | MC 64 | OURS |
|---|---|---|
| (92K VERTICES) | (5.6K VERTICES) | (4.5K VERTICES) |

Figure 12: In a *face-to-face* comparison, we examine marching cubes with 256 and 64 samplings, denoted as MC 256 and MC 64, respectively, and our method. Notably, MC 64 struggles to accurately represent pointy areas of meshes due to its coarse sampling. In contrast, our approach has an advantage in this aspect, as we analytically extract the mesh from the decision boundary apex points of networks. However, it is worth noting that our adaptive sampling introduces a disadvantage where some vertices are in close proximity; specifically, we observed that 10.1% of vertices are within 1e-3 proximity to others. We believe that optimizing the mesh can significantly improve the parsimony of the vertex count compared with marching cubes.

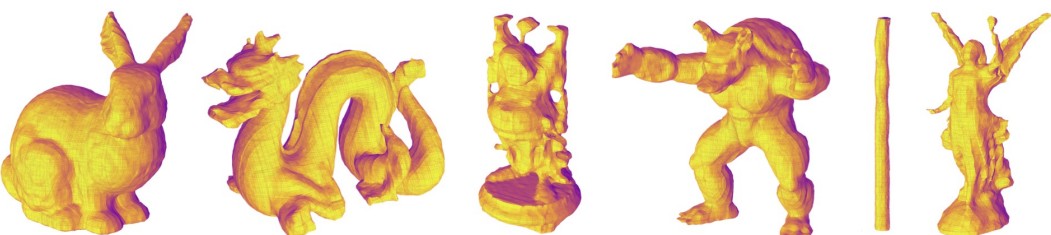

Figure 13: The extract mesh visualizations of the six from the Stanford 3D Scanning repository using the *Large* models. From left, Bunny, Dragon, Happy Buddha, Armadillo, Dril Bit, and Lucy.

