# OpenReview forum: "Polyhedral Complex Derivation from Piecewise Trilinear Networks"
_NeurIPS.cc/2024/Conference — NeurIPS 2024 poster_

### Official Review · Reviewer_1oW4 · 2024-07-09

**Soundness:** 2
**Presentation:** 1
**Contribution:** 2
**Rating:** 5
**Confidence:** 3

**Summary:**

The paper proposes a theoretical link between the tropical algebraic interpretation of ReLU-based neural networks and surfaces represented by signed distance functions trained on those networks. Based on that theoretical interpretation, a surface extraction algorithm is proposed.

**Strengths:**

(1) The mathematical notation is good overall, with good choices for the symbols and their use.

(2) The theoretical framework is interesting and it is a good application of tropical algebra.

**Weaknesses:**

(1) I do not like the current presentation of the paper. I understand that it is theoretical, but its focus on lower abstraction details from the very beginning makes it a niche paper in NeurIPS. For example, the Experiments section focus on the details of the empirical settings, moving the actual results to an Appendix. Another example is that there is little effort in providing an overview of the method. Overall this is the core design of the paper: focus on the definitions and details before properly motivating them. In its current form, this paper is more aligned with an applied mathematics journal. This is the core reason of my current rating and fixing that problem is not possible without rewriting the paper. However, I think the theoretical framework is sound so I am not recommending rejection. All other following weaknesses derive from this presentation problem.

(2) This paper needs graphical intuitions to assist unexperienced readers. For example, showing the plot of a simple tropical polynomial, with the hypersurface points highlighted, would give a good intuition of the rationale behind Definition 3.1 (i.e. the linking points between each piece-wise linear region must be where two monomials are equal). If space for the images is a problem, I would advise to include them in the supplementary material, with references in the main text. This more didactical approach would increase the public that could appreciate the paper.

(3) Another example is the subdivision scheme. Showing an schematic image of the subdivision would help understanding.

(4) I also think a better overview image is needed. Figure 1 does not make a good overview because its abstraction level is too high. A better image should refer to core symbols in the definitions. Not having a proper overview makes understanding the motivations behind some definitions more difficult. For example, $\psi$ and $H$ are defined in Definition 4.1, and I could not find the motive for them in the main text. They are only used at Proposition D.2, at the Appendix. This kind of graphical intuition is even more important in a geometrical problem, which is the case of this paper.

(5) The extracted surfaces have noise and discretization problems, probably because of the grid-based approach and interpolation. Grid-based methods are known to introduce discretization problems (Neural Geometric Level of Detail - NGLOD, Instant Neural Geometric Primitives - Instant-NGP).

(6) The evaluation is lacking. The paper only compares against classic marching cubes. Being a classic algorithm, several improvements were introduced by more recent papers over the years.

**Questions:**

(1) Typo in line 426: $x^3 + 2xy + y^4$ should be $x^3 \oplus 2xy \oplus y^4$. This is important because an unexperienced reader may get stuck in there.

(2) I believe the notation $v_j^{(i)}(x)$ in Proposition 3.3 is a little bit ambiguous. It is not immediately obvious that the $j$-th component is from the result of the layer computation $v^{(i)}(x)$ or if it is the $j$-th weight in layer $(i)$. In the second case $v_j^{(i)}(x)$ would have a weird interpretation, but I can see readers being stuck in there. Changing the notation probably would have consequences, but at least saying that the component $j$ is from the resulting vector $v^{(i)}(x)$ in the commentary that follows the definition would avoid that ambiguity.

(3) Also in Proposition 3.3 I would add an intuition commentary afterwards saying that $v_j^{(i)}(x)$ is a tropical polynomial, thus $v_j^{(i)}(x) = 0$ is an hyperplane composed of the points where two monomials are equal, intuitively connecting Proposition 3.3 with Definition 3.1.

**Limitations:**

Yes, section 7 describes limitations.

---

> ### Author Rebuttal · Authors · 2024-07-31
>
> Thank you for your thorough review and constructive feedback. We appreciate your recognition of the theoretical soundness of our framework. We would like to address your concerns regarding the presentation of our paper.
>
> **W1. I do not like the current presentation of the paper.**
>
> - We understand your concern about the paper's focus on lower abstraction details from the beginning. Our intent was to provide a solid theoretical foundation before delving into the empirical results, as we believe this balance is crucial for a comprehensive understanding. We are not simplpy "moving the actual results to an Appendix." The main results are demonstrated in Table 1 and Figure 2, showcasing the improved Chamfer efficiency across multiple benchmarks for six objects and both small and large models. Tables 2 to 8 in the Appendix provide detailed numbers and are organized separately for readers who may wish to examine the specifics.
>
> **W2. This paper needs graphical intuitions to assist unexperienced readers.**
>
> - Our new **Figure B in the rebuttal PDF** could enhance clarity as you requested. It visualizes the regions of a tropical polynomial for inexperienced readers.
>
> **W3. Another example is the subdivision scheme.**
>
> - Although we have illustrated our curved edge subdivision scheme in Figure 1 (2a-c), **Figure C in the rebuttal PDF** shows a schematic of the subdivision progress, as you suggested. We hope this feedback may resolve your concerns.
>
> **W4. I also think a better overview image is needed... For example, $\psi$ and $\mathrm{H}$ are defined in Definition 4.1, and I could not find the motive for them in the main text.**
>
> - In Definition 4.1, $\psi$ represents trilinear interpolation, and $\mathrm{H}$ refers to the target features to be interpolated. In Definition 4.3, we describe $\tau$ as "using the trilinear interpolation of spatially nearby learnable vectors on a three-dimensional grid." HashGrid (Muller et al., 2022) assigns features from hash table entries to $\mathrm{H}$ through hashing, with weights based on their relative positions in the grid. Thus, when $\tau$ is instantiated with HashGrid, $\psi$ in Equation 5 corresponds to $\tau$, where $\mathrm{w}$ represents a position in a normalized unit grid and $\mathrm{H}$ consists of the hashed features.
> - Although HashGrid, which won the best paper award at SIGGRAPH'22, is well-known, we may have assumed readers would be familiar with its formal definition (our bad.)
> - Based on your suggestion, we will consider enhancing our manuscript with textual and graphical explanations. Meanwhile, we recommend you refer to Figure 3 in Muller et al. (2022) for a visual explanation of their use of trilinear interpolation.
>
> **W5. Discretization problems, probably because of the grid-based approach and interpolation.**
> - For mesh extraction, we would like to note that sampling-based methods, e.g., MC, severely suffer from discretization, especially in low-resolution regimes. For implicit surface learning, while sinusoidal positional embedding is also known for attenuating the spectral bias, it suffers slow training/rendering time, necessitating heavy density/SDF networks compared to HashGrid. For ours, one of the possible solutions is carefully adjusting multi-resolution levels and resolution growing factors to spread multi-resolution grids more evenly.
>
> **W6. The evaluation is lacking. The paper only compares against classic marching cubes. Being a classic algorithm, several improvements were introduced by more recent papers over the years.**
>
> - We are happy to supplement on this matter. We provided experiments to compare with MT, NDC, and using a mesh simplification, QEM (Quadric Error Metrics; Garland & Heckbert, 1997). For details, please refer to our global responses **G.2** and **G.3** and the corresponding new **Figures A, D, E, and F in the rebuttal PDF**. We present both qualitative and quantitative results to highlight the advantages of our method.
> - The "several improvements over the years" appear to be tailored to each method's specific niche, each with its own pros and cons (refer to G.2). In our benchmarks, MC, utilizing our modernized library PyMCubes, outperforms others in terms of efficiency, establishing it as the de facto method.
>
> **Q1. Typo in line 426**
>
> - Actually, this is not typo; but it embraces ambiguity. We want to say the *classical* polynomial $x^3 + 2xy + y^4$ would be rewritten for a *tropical* polynomial by naively replacing + with $\oplus$ and $\cdot$ with $\odot$, respectively: $x^3 + 2xy + y^4$ -> $x \odot x \odot x \oplus 2 \odot x \odot y \oplus y \odot y \odot y \odot y$. After this, we rewrite by the definitions of tropical operations as $\max(3x, x + y + 2, 4y)$. This example (ours with max notation) comes from the Wikipedia (https://en.wikipedia.org/wiki/Tropical_geometry) and we believe this is a convention how they refer the *classical*/*tropical* polynomials. But, we would like to clarify that in revision.
>
> **Q2. Notation in Proposition 3.3**
>
> - Yes, the component j is from the resulting vector $v_j^{(i)}(x)$. We stated "the subscript j denotes the j-th element (neuron index; the j-th element in the intermediate output), assuming the hidden size of neural networks is H." Since we define j in \[1, H\] we thought readers would follow. Your suggestion is valuable and will be incorporated into the update.
>
> **Q3. Also in Proposition 3.3 I would add an intuition commentary afterwards saying that...**
>
> Great point! We appreciate that.

---

> > ### Comment · Reviewer_1oW4 · 2024-08-07
> >
> > Thank you for considering the review and the efforts to answer the comments. I have no further questions.

---

> > > ### Author Response · Authors · 2024-08-08
> > >
> > > Thank you for your prompt response.
> > >
> > > We humbly ask that you please inform us if you have any remaining major issues that could affect the rating so we can address them during the author-reviewer discussion period. In particular, we have included Figures B and C in the rebuttal PDF to address your main concerns.

---

### Official Review · Reviewer_FmNY · 2024-07-12

**Soundness:** 2
**Presentation:** 2
**Contribution:** 3
**Rating:** 5
**Confidence:** 4

**Summary:**

The paper extends the idea of extracting the polyhedral complex from ReLU networks to extracting a more general cell complex from ReLU networks with a positional encoding, specifically, trilinear encodings, such as HashGrid and TensoRF, commonly used in neural field representations of geometries or scenes.
It first theoretically extends the concept of piecewise linear to piecewise trilinear networks. It then proposes a modified *curved* edge subdivision algorithm to extract a polyhedral mesh approximating the neural implicit surface. The key assumption governing the approximation error is the eikonal loss, which ensures that the isosurfaces of the trilinear functions are close to planar.
The algorithm is validated on several shapes qualitatively and quantitatively, comparing to marching cubes in terms of the mesh size, accuracy, and time. It demonstrates sparser meshes at an increased computational cost.

**Strengths:**

The attempt to generalize the insights from piecewise linear networks to ReLU networks with positional encodings is well motivated and holds a lot of promise. To the best of my knowledge this is the first attempt at such a characterization as well as a practical implementation. The theoretical aspect of the work is in my view the most clear contribution (however, I could not fully verify the details).

**Weaknesses:**

Despite the method being able to handle volumetric complex of the input space, the the work only considers the "skeletonized" complex, i.e., the boundary mesh. At least briefly describing the former would make for a more rounded story. Instead, the skeletonized method is framed as a potential alternative to marching cubes, which unfortunately is not convincing from the results. The only demonstrated advantage is in terms of the mesh size. This argument would be more convincing given a comparison to marching cubes with some basic mesh post-processing. The work would also benefit from comparing the same procedure to piecewise linear networks and the underlying edge subdivision method in terms of training time, extraction time, mesh size, and accuracy.

Similarly, while the method is evaluated for different positional encodings, the neural network size remains fixed (3x16) making it hard to judge how the method scales in practice.

The topic and the proposed method are nuanced and technical, so it does not help that there are many imprecise statements, including in theorems, and poor phrasing, which render the manuscript hard to follow (see questions and suggestions below). The authors also overlook the chance to help the reader with more visual aids.

Overall, I think the work has great potential which is not fully realized and publication-ready due to the unrefined presentation and somewhat uninspiring application.

**Questions:**

- In Theorem 4.5, is it true that $\tau$ is a trilinear function? While $\tau$ is defined elsewhere in 4.3, this assumption is missing in the theorem itself which is confusing. You should also clarify in the manuscript how trilinearity is used in the proof, e.g. in (41) - (43).
- WIth that, could you clarify Theorem 4.5? My best understanding is that it states that the only trilinear function which satisfies the eikonal equation is a plane, but a counterexample to this is easy to conceive.
- In Line 221, do you perhaps mean $ |\tilde{\nu}(\mathbf{x})| \leq \epsilon$? Otherwise, you select the edges of the sub-level set, instead of the level set.
- Why do you use the $256^3$ marching cubes mesh as a ground-truth instead of using the input mesh on whose SDF you train?
- Can you comment on the extraction of the volumetric mesh? Is my understanding correct that this is what you have before the skeletonization step? Does this and your method more broadly offer a way to explain the success of trilinear positional encodings in terms of the underlying complex?

**Limitations:**

The limitations paragraph focuses solely on the runtime of the method. The section would benefit from reiterating and discussing the reliance on the eikonal constraint, which is known to be hard to satisfy globally. I would also urge the authors to consider discussing the limitations of the experiments as described in the weaknesses.


**Errors and suggestions**

- The motivation of discussing the normals in 4.3 is unclear.
- All the left-hand terms in 40-43 should be squared. The rhs in (40) is the square of the gradient norm, not the gradient norm.
- The work would benefit from a discuss on the size of the NN and the encoding for practical applications. To my surprise, [12] uses similarly sized networks to your 3x16 with decent performance.

- Line 19: "polyhedral" -> 'polyhedron"
- Line 50: "This provides a theoretical exposition of the eikonal constraint" is unclear
- Line 59: you should give some examples of where marching cubes are used in a learning pipeline and explain what you mean by redundancy (probably flat regions which are described by many triangles?)
- Equation 2: I would not recommend overloading the notation for $d$
- Definition 3.2: what do you mean by "without loss of generality" when defining the NN?
- Line 109: "subscription" -> "subscript"
- Line 110: "for rather discussions as an SDF" is entirely unclear
- Lines 112-114 are very unclear
- Line 127: "subdivising" -> "subdividing"
- Definition 4.1: what is $\psi$? You should also define/explain $F$ before using it. Why is the definition given for general $D$ if this is a trilinear interpolation?
- Line 203: "also the" -> "also due to the" ?
- Line 248: ", similarly" is not needed
- Line 259: Can you be more precise with the objective? The goal is to get a boundary mesh of the zero-set of the SDF.
- Line 262: reformulate to use a verb
- Line 276: CE is not defined in the main text.
- Line 294: "plenary" -> "planarity"?
- Line 560: do not write "by the way"
- Overall, please check the text with a spell and/or style checker

---

> ### Author Rebuttal · Authors · 2024-07-31
>
> We sincerely appreciate your efforts and the detailed comments you provided. We are particularly pleased that the theoretical aspects of the work are considered a significant contribution.
>
> **At least briefly describing the former would make for a more rounded story.**
>
> - In Sec. 1, we introduce mesh extraction from Continuous Piecewise Affine (CPWA) functions, which are limited by spectral bias and typically use nonlinear positional encodings for this limit. Consequently, applying CPWA mesh extraction techniques becomes challenging. In Sec. 2, we discuss how many previous works face significant challenges due to the computational cost, with exponentially growing linear regions. Linear regions grow polynomially with the hidden size and exponentially with the number of layers (Zhang et al., 2018; Serra et al., 2018). These issues were our key motivations.
> - The elaboration on your "handling volumetric complex" would help provide specific feedback on that matter.
> - Also, could you refer to **our global response G.1** for "Why does the chamfer efficiency matter?"
>
> **This argument would be more convincing given a comparison to marching cubes with some basic mesh post-processing.**
>
> - We provided supplementary experiments to compare with MT, NDC, and using a mesh simplification, QEM (Quadric Error Metrics; Garland & Heckbert, 1997). Please refer to our global responses **G.2** and **G.3** and the corresponding new **Figures A, D, E, and F in the rebuttal PDF**. We present qualitative and quantitative results on them.
>
> **The work would also benefit from comparing the same procedure to piecewise linear networks...**
>
> - We are afraid you misunderstand. Our motivation stems from the issue of spectral bias (Tancik et al., 2020) in implicit neural surface learning methods, which necessitates the use of positional encoding, such as HashGrid (Muller et al., 2022). Therefore, exploring mesh extraction from these types of networks is a key focus of our work. Comparing our method to piecewise linear networks (first, it is hard to learn surface naively due to the spectral bias.) could be distracting and misguided, as our goal is not to argue that HashGrid improves quality in addressing spectral bias -- this has already been well-discussed and established in previous works.
>
> **The neural network size remains fixed (3x16)?**
>
> - Networks using HashGrid (Muller et al., 2022) typically have **a small hidden size of 64 and only one hidden layer**, even in implicit neural surface learning, NeuS2 (Wang et al., 2023). This is because HashGrid contains a large number of learnable parameters in its hash tables, specifically (feature size 2) x (# of entries 2^19 = 524K) x (multi-resolution levels 4). In our experiments, we used a reasonable network capacity. For clarification, the input size of 3 is mapped to (levels 4 x features 2 = 8) by HashGrid, and then the decoding network maps 8 -> 16 -> 16 -> 1 with 3 mapping layers (as described in Sec. 6).
>
> **Q1. In Theorem 4.5, is it true that \tau is a trilinear function?**
>
> - Yes. We defined $\tau$ as a trilinear function in Definition 4.3 in Sec. 4.1 (the previous section). In Definition 4.1, $\psi$ represents trilinear interpolation. In Definition 4.3, we describe $\tau$ as "using the trilinear interpolation of spatially nearby learnable vectors on a three-dimensional grid." relating to $\psi$ for instantiation. We will revise to have a formal definition of $\tau$, enhancing clarity.
>
> **Q2. Could you clarify Theorem 4.5? My best understanding is that it states that the only trilinear function that satisfies the eikonal equation is a plane...**
>
> - Theorem 4.5 states that a trilinear function (within a piecewise trilinear region), which satisfies the eikonal equation, represents a plane. We do not argue that **only** trilinear function which satisfies the eikonal equation is a plane. We choose the trilinear networks using HashGrid since 1) it effectively handles the spectral bias, 2) it is one of the widely-used methods in NeRFs, and 3) it has a planarity opportunity using the eikonal constraint. Notice that we provided detailed proof in Theorem D.5.
>
> **Q3. In Line 221, do you perhaps mean $\|\tilde{v}(\mathrm{x})\| \le \epsilon$?**
>
> - Yes. Thank you for pointing out the typo.
>
> **Q4. Why do you use the 256^3 marching cubes mesh as a ground-truth instead of using the input mesh on whose SDF you train?**
>
> - To eliminate Bayes error from underfitting. This work is solely focused on the extraction itself, not on surface learning. Notice that we used $512^3$ for large models, and confirm that it were sufficiently high resolution for meaningful evaluation.
>
> **Q5. Can you comment on the extraction of the volumetric mesh?**
>
> - The question seems unclear to us; but we would like to explain that before the skeletonization, the initial volumes (a cube of interest) are partitioned by grid marks (grid planes) and folded hypersurfaces. In tropical geometry, these are known as tropical hypersurfaces (Itenberg et al., 2009), which are sets of points where the function is not linear (or zero). Before the skeletonization step, we identify all non-linear boundaries and select the zero-set vertices and edges for our efficient extraction. **Please refer to Fig. C in the rebuttal PDF** for a visual aid.
>
> **Limitation**
>
> - Thank you for pointing that out. We want to draw your attention to Appendix B.1, which discusses "Satisfying the eikonal equation and approximation." In this section, we examine the effectiveness of HashGrid in satisfying the constraint through the piecewise allocation of hash table entries and their locality. We will integrate this discussion into the Limitations section.
>
>
> **Errors and suggestions**
>
> - We sincerely appreciate your effort in providing a detailed list of errors and suggestions. We will incorporate your feedback to enhance the clarity of our work. Note that a discussion of HashGrid ( \[12\]) has been addressed above (model size).

---

> ### Author Response · Authors · 2024-08-08
>
> *Below are our comments on your questions, demonstrating our commitment to clarifying our paper.*
>
> ---
>
> **The motivation of discussing the normals in 4.3 is unclear.**
>
> - To define a face, we use a list of vertex indices. To determine the normal vector of the face (considering the ambiguity of two opposite directions), we implicitly rely on the order of the vertex indices as described in the paper.
>
> **Line 50: "This provides a theoretical exposition of the eikonal constraint" is unclear**
>
> - This indicates Thm 4.5 and Coro. 4.6 which stated "that within the trilinear region, the hypersurface transforms into a plane" under the eikonal constraint.
>
> **Line 59: you should give some examples of where marching cubes are used in a learning pipeline and explain what you mean by redundancy (probably flat regions which are described by many triangles?)**
>
> - MeshSDF (Remelli et al., 2020) will be the example. Yes, your answer is right.
>
> **Line 110: "for rather discussions as an SDF" is entirely unclear**
>
> - An SDF has a single scalar output as a definition. In the manuscript, we later discuss the networks as an SDF in Sec. 4, specifically Def. 4.3. (Line 175). We will make sure to clarify this as you asked.
>
> **Lines 112-114 are very unclear**
>
> - We hope **Fig. C (c) in the rebuttal PDF** helps your understanding. Our method performs a single pass over neurons to find tropical hypersurface (non-linear boundaries; refer to **Fig. B in the rebuttal PDF**).
>
> **Definition 4.1: what is $\psi$? You should also define/explain $F$ before using it.**
>
> - In Def. 4.1, $\psi$ is a trilinear interpolation function given weight $\mathrm{w}$ and features $\mathrm{H}$. This is implemented using the HahsGrid as $\tau$ in Def. 4.3. In Lines 172-173, we explained that "they transform the input coordinate $\mathrm{x}$ to $\mathrm{w}$ with a multi-resolution scale and find a relative position within a grid cell." The features $\mathrm{H}$ are the hash table entries derived from the hashing function, where the input to the hash function is a given grid vertex and the output is the index in the hash table. We will provide a clear explanation in the revision.
>
> - $F$ is the output dimension of $\tau$, positional encoding module. (Line 170)
>
> **Line 259: Can you be more precise with the objective? The goal is to get a boundary mesh of the zero-set of the SDF.**
>
> - By the definition of SDF, the zero-set of the SDF naturally indicates surfaces. But we respect your concern and will revise it.

---

> ### Author Response · Authors · 2024-08-12
>
> Dear Reviewer FmNY,
>
> As we approach the end of the reviewer-author discussion period, we want to ensure that our feedback has sufficiently addressed your major concerns. We recognize the importance of this opportunity to address the reviewers' raised concerns, and we are committed to providing any necessary clarifications. Please let us know if you have further questions or need additional clarification.
>
> Best regards,
>
> On behalf of all authors

---

> ### Comment · Reviewer_FmNY · 2024-08-12
>
> I thank the authors for their detailed response. While some of my minor concerns are addressed, the presentation and clarity remain my main concern.
>
>
> **Figure C**
> I think this schematic is very misleading. The final decision boundary (0 level-set, red) is not composed of some intermediate neuron decision boundaries in the way you show. I would strongly encourage you to visualize some intermediate steps from your method accurately depicting a realistic subdivision.
>
> If this is not possible in the available time due to how your code is, a simple way to visualize the partition is to pass the coordinates of the pixels through the NN and plot the pre-activations of the ReLU neurons (e.g. Figures 2 or 7 in Hanin, B. et al, Deep ReLU networks have surprisingly few activation patterns, NeurIPS 2019.)
>
>
> **MC + QED**
> I appreciate the additional MC+QED experiment, but I slightly question the setup - this is reported for the _Large_ model, which is mainly discussed in the Appendix (Table 6). There, your method has better CD than any MC, so of course QEM will perform better on your mesh. I would appreciate if you could report MC+QEM for the _Small_ (Table 1) which is used in the main paper and where finer MC meshes (128 or 196) have better CD but worse vertex counts.
>
>
> **Theorem 5**
> Could you please clarify Line 559 in the proof? Do you perhaps mean for _all_ $\mathbf{x}$?
>
>
> **Model size**
> My criticism is not that the size itself it small, but rather that the method is assessed just for a _single size_. E.g. Fig. 10 of instantNGP [12] reports multiple widths and depths of the MLP, so it would be reassuring to see how the quality/runtime behaves for models of size other than 3x16.
>
>
> **Comparison to CPWA**
> I understand the motivation of the spectral bias and positional encoding. However, because you are treating this topic from the perspective of surface meshes, contrasting the meshes obtained with and without an encoding is in my view a worthwhile consideration. Describing and explaining the qualitative and quantitative differences between these meshes could add to a unique interpretation of positional encodings and strengthen the unique aspect of your work. The whole point of increasing the spectral bias is to represent higher frequency components - are these present in the mesh relative to the plain ReLU NN? Training a Standford bunny with ReLU certainly isn't hard (the method you reference in [4] and the code therein already provides this). I do not expect the authors to perform the experiments and the analysis in the short available time and this is not critical for my current assessment, but as a discussion point, I do maintain that this could add to your unique work overall.
>
>
> **Q4**
> Thank you, this clarifies my concern on the ground-truth.

---

> ### Author Response · Authors · 2024-08-12
>
> **Figure C**
>
> We would like to clarify a potential misunderstanding. While Figure C is a schematic figure, we believe it accurately illustrates the characteristics of our method. The final decision boundary (0 level-set, shown in red) *may exclude* some intermediate neuron decision boundaries, as demonstrated. In Definition 3.1 (p.3), we introduced the concept of the **tropical** hypersurface, corresponding to the nonlinear boundaries formed by all neurons (including an SDF). We then referred to Proposition 3.3 (Decision boundary of neural networks), which stated that the decision boundary is a tropical hypersurface; however, the reverse is not true. Not all points on the tropical hypersurface constitute the decision boundary at the zero level-set. We formally denoted this relationship as $\mathcal{B} \subseteq ...$, rather than as $\mathcal{B} = ...$ in Proposition 3.3. And, in Lines 112-113, we stated that "the preactivations, $\nu_j^{(i)}$ are the *candidates* shaping the decision boundary." For more rigorous explanations on this matter, please refer to Proposition 6.1. of Zhang et al. (ICML 2018) "Tropical Geometry of Deep Neural Networks."
>
> Notice that there is one more detail. The bottom line of the bunny is from a grid line, not from neuron's folded hypersurface. Geometrically, the grids of the HashGrid is non-linear boundaries which efficiently divided the Euclidean space into a lattice structure. To achieve this via only neurons, we need more neurons, which may explain why HashGrid (Muller et al., 2022) does not need many neurons or layers in decoding networks.
>
> Your constructive feedback has highlighted areas that may lead to misunderstandings, and we are fully committed to improving the clarity of our presentation.
>
> Thank you for suggesting the figure from Hanin & Rolnick (2019). We will consider visualizing our concepts using this reference. Sadly, we cannot update the rebuttal PDF in the reviewer-author discussion period.
>
>
> **MC + QED**
>
> Unfortunately, the *Small* models have coarser grid intervals in the HashGrid, which puts them at a disadvantage compared to the *Large* models in terms of our analytical approximation method. (Denser grid intervals result in smaller errors when approximating curved hypersurfaces.) Additionally, using the *Small* models is less representative of real-world scenarios than the *Large* models.
>
>
> **Theorem 5**
>
> Yes, that appears to be an unintended typo. We will correct it.
>
> **Model Size**
>
> Referring to `3x16` could be misleading. Our model has three layers, structured as 3 (coordinates) -> 8 (hash grid) -> 16 -> 16 -> 1.
>
> One limitation of this ablation study is that model size impacts fitting, affecting the GT mesh (MC256 for the pseudo-GT). However, the major model capacity comes from the hash tables (explained above in the initial author feedback). Therefore, the changes in CD are small. For completeness, we provide the following tables:
>
> For the Bunny Small models,
>
> | Layers | Width | Vertices    | CD (x 1e-6) | Time (sec) | Note       |
> |--------|-------|-------------|-------------|------------|------------|
> | 3      | 4     | 4547        | 666         | 0.12       |            |
> | 3      | 8     | 4661        | 750         | 0.19       |            |
> | 3      | 16    | 4628        | 724         | 0.65       | Ours       |
> | 3      | 32    | 4567        | 754         | 0.57       |            |
> | 3      | 64    | 5588        | 759         | 2.86       |            |
> | 2      | 4     | 4565        | 738         | 0.11       |            |
> | 2      | 8     | 4863        | 768         | 0.13       |            |
> | 2      | 16    | 4566        | 734         | 0.12       |            |
> | 2      | 32    | 4577        | 754         | 0.22       |            |
> | 2      | 64    | 4611        | 739         | 0.50       | InstantNGP |
>
> Notice that the model with layer=3 and width=64 took 2.86, because of the large number of intermediate (not final) vertices. The time complexity is more sensitive to the number of intermediate vertices than to the number of layers or the width.
>
> W.r.t. the number of layers:
>
> | Layers | Width | Vertices    | CD (x 1e-6) | Time (sec) | Note       |
> |--------|-------|-------------|-------------|------------|------------|
> | 2      | 16    | 4566        | 734         | 0.12       |            |
> | 3      | 16    | 4628        | 724         | 0.65       | Ours       |
> | 4      | 16    | 4569        | 759         | 0.51       |            |
> | 5      | 16    | 4579        | 749         | 0.59       |            |
>
> The number of layers has a limited impact on runtime, especially for 3 to 5.
>
> **Comparison to CPWA**
>
> I appreciate your insight into how it could enhance the overall work. However, we have found that the experiments require a major code revision due to integration issues with HashGrid. Although, we will consider this in our revision.

---

> ### Comment · Reviewer_FmNY · 2024-08-13
>
> I thank the authors for the clarification. I appreciate the additional table reporting the runtimes for different model sizes. I would encourage you to include this in the appendix.
>
> Given that some of my concerns have been addressed and given the overall commitment of the authors to improve the work, I will increase my rating to borderline accept.
>
> However, my main concern remains with potentially publishing misleading information. I urge the authors to take time to consider the subdivision process and redo Figure C for the final version. Currently, Figure C *is* wrong - in the generic case, the final decision boundary does not overlap with intermediate neuron decision boundaries. Each neuron in a ReLU network subdivides some existing pieces. The composition with the positional encoding changes the shape of these pieces but not the fact that these are still subdivided in a generic manner.
> Please consult existing works describing this, e.g.
> - Figure 4 in Humayun et al. SplineCam: Exact Visualization and Characterization of Deep Network Geometry and Decision Boundaries. CVPR 2023
> - Figure 3 in Jordan et al. Provable Certificates for Adversarial Examples: Fitting a Ball in the Union of Polytopes. NeurIPS 2019.
>
> You can also verify this yourself using the approach I described previously.
>
>
> Similarly, you cannot simply state that _Small_ performs worse so you are just not going to report the results.

---

### Official Review · Reviewer_1zQb · 2024-07-12

**Soundness:** 3
**Presentation:** 2
**Contribution:** 2
**Rating:** 5
**Confidence:** 1

**Summary:**

The paper presents a theoretical and practical framework to extract meshes from piecewise trilinerar networks using hash grid representation. They further show that hypersurfaces within the trilinear regions become planes under the eikonal constraint. A method for approximating the intersection between 3 hypersurfaces is proposed (approximated as 2 hypersurfaces and a diagonal). Evaluation is performed against marching cubes resolutions using chamfer distance, chamfer efficiency and angular distance.

**Strengths:**

- Thorough theoretical analysis
- Proof of concept practical evaluations show improvement over naive marching cubes at similar complexity of meshes

**Weaknesses:**

- While there are some improvements over marching cubes, the improvements are not quite substantial, and come at the cost of a higher runtimes
- No comparisons to any alternate approaches over marching cubes are provided -- e.g. marching cubes + QEM decimation etc.
- No rigorous qualititative comparisons are provided, which could be important since chamfer does not always capture the shape quality

**Questions:**

Would be nice if the authors can add some qualitative comparisons and also against more than just marching cubes. Further current evaluation is only performed on a handful of meshes, would be nice if an evaluation on a bigger dataset can be performed

**Limitations:**

Yes

---

> ### Author Rebuttal · Authors · 2024-07-31
>
> Thank you for your thorough review and constructive feedback. We explain our rationale for the evaluation and additional note for the angular distance we used in quantitative evaluations.
>
> **W1. While there are some improvements over marching cubes, the improvements are not quite substantial, and come at the cost of a higher runtimes**
>
> - Our method is a white-box approach that identifies apex vertices with a reasonable eikonal constraint, showing the improved Chamfer efficiency (Please refer to **our global response G.1** on this matter) across multiple benchmarks for six objects and both small and large models. After submission, by utilizing PyTorch's `masked_scatter_`function to identify intersecting edges and others, we significantly reduced the processing time from 53.5 seconds to approximately 3 seconds for the Dragon (Large) model, achieving similar time spent comparable to MC. This result highlights the potential for further improvements in efficient implementation. We have also provided a theoretical analysis of the time complexity (see Appendix B), which suggests that our approach is theoretically optimal. The code will be publicly released upon acceptance.
>
>
> **W2. No comparisons to any alternate approaches over marching cubes are provided -- e.g. marching cubes + QEM decimation etc.**
>
> - Thank you for your suggestions. Please refer to our **global responses G.2 and G.3** and the corresponding **new Figures A, D, E, and F in the rebuttal PDF**.  We provide supplementary experiments to compare with MT, NDC, and the QEM scenario.
>
> **W3. No rigorous qualitative comparisons are provided, which could be important since chamfer does not always capture the shape quality**
>
> - **Please refer to our Figures A & F in the rebuttal PDF, which show detailed extracted vertices and edges and normal errors for a visual explanation.** **Fig. A provides rigorous visualization with enlarged meshes and normal directions are visualized with plasma colors.**
> In addition to the Chamfer distance, we present the angular distance in Table 5 in the Appendix. This measurement, derived from the extracted normal vector, a direction indicated by the normalized Jacobian of the SDF function, assesses how much the normal vectors deviate from the ground truth (detailed in Sec. 6). As anticipated, our method effectively estimates the faces using a relatively small number of vertices compared to the sampling-based approach. This trend is consistently observed across various objects. We believe this supplements the limited aspect of the chamfer distance.

---

> > ### Author Response · Authors · 2024-08-13
> >
> > Dear Reviewer 1zQb, as we approach the conclusion of the reviewer-author discussion period, do you have any remaining questions, or have your concerns been fully addressed? Your confirmation and reconsideration of your rating would be greatly appreciated.

---

> > > ### Comment · Reviewer_1zQb · 2024-08-13
> > >
> > > Dear authors, thank you for you rebuttal. I appreciate the implementation improvement which greatly improves the time taken for meshing by the approach. However, I still think an evaluation an evaluation on a bigger set of meshes can strengthen the statistical significance of the quantitative evaluations, since the current evaluation is performed on very limited set of meshes. Further, visually I still find the benefits of the approach not too substantial, e.g. in Fig. F, it is mentioned that the improved normals would help lighting, but the reader has no idea how significant this improvement is going to be. We see that QEM helps both proposed and MC, I still wonder (a) how "MC + QEM" vs "Ours + QEM" looks qualitatively for smaller and larger meshes, and
> > > (b) how "MC + QEM" compares against the proposed approach both qualitatively and quantitatively, to see how helpful the proposed approach is.
> > >
> > > I will keep my initial rating of borderline accept

---

### Official Review · Reviewer_MSAr · 2024-07-13

**Soundness:** 3
**Presentation:** 2
**Contribution:** 2
**Rating:** 5
**Confidence:** 2

**Summary:**

The work addresses the problem of exact iso-surface extraction given an implicit surface represented by a neural network. The idea behind this line of work is to closely analyze the architecture of the neural network to efficiently extract the true (or at least an approximate of) learned iso-surface. Previous work titled 'Polyhedral Complex Extraction from ReLU Networks using Edge Subdivision' develops the analysis for ReLU networks. However, many newer works use local latent codes to achieve better reconstruction quality. These latent codes are usually obtained via trilinear interpolation from a grid of learnable latent codes. The previous method does not trivially apply to iso-surface extraction in this case. The current work continues the analysis for ReLU networks that now have trilinear positional encoding as input. The paper first shows that under the eikonal constraint, trilinear regions decode into flat planes. Then, it derives a polynomial defining the intersection of two hypersurfaces. With this, the paper suggests an algorithm for zero-level set extraction given the trained neural network and the positional encoding.

**Strengths:**

* The paper extends previous work from “Polyhedral Complex Extraction from ReLU Networks using Edge Subdivision” to architectures with positional encoding (PE) implemented via trilinear interpolation. This is a very reasonable extension, as many implicit representation methods in practice use grids of local embeddings.
* Theorem 4.5 (and Corollary 4.6), which prove that the trilinear regions of PE decode into flat planes under the eikonal constraint, seem surprising. If true, this is a great insight into the performance of these models.
* The suggested algorithm can extract a good approximation of the learned surface and matches the accuracy of Marching Cubes at much smaller resolutions.
* The paper is self-contained and presents interesting insights into what neural implicit representation methods are capable of reconstructing.

**Weaknesses:**

Here are the main concerns I found in the paper in order of importance (with the first being the most important):

* I found the paper very hard to understand. I am quite familiar with neural implicit representations (though not with Polyhedral Complex Extraction), but it still took me at least three full passes before I started understanding the general idea of the paper. The paper is self-contained and formal, which is great, but I would consider simplifying the language for the sake of understandability.

* The work is positioned as a better way of iso-surface extraction, yet there is no visualization of extracted meshes in the main paper. It is very hard to say if the mesh extracted with the proposed method is actually better than the ones extracted with Marching Cubes. In my view, MC64 seems to be actually closer to MC256 than the mesh extracted with 'ours' in Figure 7. A much more detailed visual comparison of extracted meshes seems necessary in the main paper given the claims.

* The method is shown to work only with relatively small neural networks: the experiments are performed on a model with three layers and a hidden size of 16, while DeepSDF, for instance, operates on eight layers, each with a width of 512. It seems that it can be used only as a source of understanding the behavior of these architectures, not as a practical method of iso-surface extraction.

* Comparison with only Marching Cubes seems very limited; it would be great to see a comparison with 'Analytic Marching' as well.

I will consider adjusting the final rating if weakness #2 and question #1 are addressed in the rebuttal

**Questions:**

* I am quite confused about the following (probably due to my limited understanding of the math in the paper): A standard ReLU network with a constant latent code as input (i.e., DeepSDF) can be seen as a trivial trilinear positional encoding (i.e., the entire domain is trilinear). Given that such networks are usually trained to satisfy the eikonal constraint, why aren’t they limited to always produce flat plane reconstructions, given Corollary 4.6?

* It seems that a natural measure of how accurately the algorithm extracts the learned iso-surface would be:

        1. Sample N points on the surface of the extracted mesh
        2. Predict SDF on each of them
        3. Calculate sum of squared SDF values
it should be 0 if we extracted the learned surface precisely. Would be interesting to see this metric. Can we confirm Corollary 4.6 with it?

**Limitations:**

The limitations are well discussed. I would also add a discussion on the depth and width of the neural networks for which the method is practical to use.

---

> ### Author Rebuttal · Authors · 2024-07-31
>
> Thank you for your detailed review and for pointing out the effective rebuttal points. We appreciate your time and effort in going through our work multiple times.
>
> **W1. Simplification of Language**
>
> We are committed to making our paper more accessible. Based on your feedback, we will revisit the language and structure to ensure that the key ideas are presented as clearly as possible. Specifically, we will: 1) Simplify complex sentences and terminology where possible. 2) Add a brief introductory section that explains Polyhedral Complex Extraction in simpler terms before delving into the technical details. Please refer to Fig. 1, and Alg. 1 & 2 in the Appendix for visual and textual aids illustrating core concepts. We believe our new **Figure B & C (rebuttal PDF)** would also help readers.
>
> **W2. Pitfall to Over-Complex**
>
> There is a *risk* that MC64 appears more similar to MC256. This is because we used small networks in this visualization to highlight the limits of MC with a limited number of vertices and edges in GT (simulating flat or high curvature/edged regions). Here, MC64 tends to oversample the vertices. When examining the nose of the rabbit in MC256, the same bright yellow color can be observed in areas where our method extracts flat facets. In contrast, MC64 oversamples around the edges, creating an over-smooth surface where it should not. For instance, when a cube is tilted at a 45-degree angle, its vertices may not be sampled as they lie within the sampling grids, leading to the creation of a multi-facet polytope. This is an innate weakness in sampling-based methods. Additionally, inaccuracies in estimating normal vectors from it will lead to incorrect light reflection outcomes. **Please refer to Figure A (rebuttal PDF)**, which shows detailed extracted vertices and edges and **Fig. F** for the normal errors in the Large models, for a visual explanation.
>
> **W3. Computational Complexity**
>
> As mentioned in Section 4.2 and detailed in Appendix B, the time complexity of our method is linear with respect to the number of vertices, making it optimal for extraction. After submission, by utilizing PyTorch's `masked_scatter_` function to identify intersecting edges and others, we significantly reduced the processing time from 53.5 seconds to approximately 3 seconds for the Dragon (Large) model, which is in between MC and MT (Marching Tetrahedra), much closer to MC. We have provided both a theoretical analysis of the time complexity (Appendix B) and practical benchmarks using the Large models to validate our findings. *The code will be publicly released upon acceptance.*
>
> Moreover, we emphasize HashGrid allows us to use small decoding networks with hash tables of large parameters.
> Networks using HashGrid (Muller et al., 2022) typically have **a small hidden size of 64 and only one hidden layer**, even in implicit neural surface learning, NeuS2 (Wang et al., 2023). This is because HashGrid contains a large number of learnable parameters in its hash tables, specifically (feature size 2) x (# of entries 2^19 = 524K) x (multi-resolution levels 4). In our experiments, we used a reasonable network capacity. For clarification, the input size of 3 is mapped to (levels 4 x features 2 = 8) by HashGrid, and then the decoding network maps 8 -> 16 -> 16 -> 1 with 3 mapping layers (as described in Sec. 6).
>
> **W4. Additional comparisons**
>
> Unfortunately, Analytic Marching is not feasible for our benchmarks due to its high time complexity. Specifically, its time complexity is $\mathcal{O}((n/2)^{2(L-1)} |\mathcal{V_P}| n^4 L^2)$, which grows exponentially with the number of layers L and the width n of the networks, where $|\mathcal{V_P}|$ represents the number of vertices per face (Section 4.1 from Lei and Jia, 2020). In contrast, our method has a linear time complexity with respect to the number of vertices, as discussed in Section 3.2, allowing it to avoid visiting every linear region exponentially growing. Instead, we would like to direct you to **our global response G.2**, which includes supplementary experiments comparing our method with MT and NDC.
>
> **Q1. Why aren’t they (DeepSDF) limited to always produce flat plane reconstructions?**
>
> As you might know, HashGrid (Muller et al., 2022) includes learnable parameters, the hash tables. The eikonal loss causes the hash entries to form a flat plane within trilinear regions. We denote the entries as $\mathrm{H}$, which are learning parameters, in Definition 4.1 in Section 4.1. For this reason, the constant latent codes cannot be applied with Corollary 4.6. However, since a standard ReLU network (with a constant latent code) is piecewise linear, it may produce flat planes in linear regions. Nonetheless, we want to emphasize the spectral bias (Tancik et al., 2020) discussed in the Introduction, which arises without appropriate positional encodings, and it serves as our motivation. As a result, DeepSDF (Park et al., 2019) may be limited to simpler shapes and suffer from this spectral bias.
>
> **Q2. Quantitative metric to validate Corollary 4.6**
>
> We appreciate the proposed metric; however, we must point out that our existing metrics already measure the suggested aspect. Firstly, using the Chamfer distance, we sample points on the surface of the extracted mesh and calculate the distance to the closest point (which is defined as the SDF value) on the ground truth mesh. This mesh is derived from the over-sampled MC, reflecting the learned surfaces of the networks. We believe this is essentially the same metric (under sufficient samplings and well-fitted SDF networks) as the one you suggested.
>
> Furthermore, we want to emphasize Equation 13 and Figure 3, which illustrate how flatness changes with respect to the weight of the eikonal loss using the flat constraints identified in the proof of Theorem D.5. We have confirmed that the eikonal loss is an effective method to ensure flatness and evaluate the extracted mesh using the Chamfer distance.

---

> > ### Author Response · Authors · 2024-08-10
> > **Follow-Up on Rebuttal Feedback**
> >
> > Dear Reviewer MSAr,
> >
> > Thank you for your thorough review and the constructive feedback you provided.
> >
> > We would like to kindly inquire whether our responses to Weakness #2 and Question #1 have addressed your concerns sufficiently, as you mentioned you might consider adjusting the final rating based on our rebuttal. Given the limited time for the author-reviewer discussion period, we would also appreciate it if you could let us know if there are any additional questions or clarifications needed.
> >
> > Thank you again for your time and effort.
> >
> > Best regards,
> >
> > On behalf of all authors

---

> > > ### Comment · Reviewer_MSAr · 2024-08-11
> > >
> > > Dear authors, thank you a lot for addressing my questions, it clarified some of my concerns.
> > >
> > > Q1: My misunderstanding was coming from treating the "trilinear region" in line 50 as the trilinear region of the PE, not a trilenar region of PE + neural network. Your response clarified it. Thank you a lot.
> > >
> > > W2 + Q2: I am still a bit confused about the metrics and visual comparisons in the paper. Even in the provided zoom-in to the nose of the Stanford bunny, it is still hard to see if the proposed method is closer to MC256. It is also hard to see the oversampling of the vertices mentioned in the rebuttal: the proposed method has only 20% fewer vertices, and the highlighted region contains the small facets along the nose line that are not present in MC64.
> > >
> > > Regarding the quantitative evaluation, I agree that in the limit of MC resolution currently used metric should converge to the one proposed in Q2. However, MC256^3 has only 4 subdivisions of each edge from MC64^3, which is still quite low resolution and introduces additional noise into the metric. Wouldn't it be more direct and transparent to directly report a sum of squares for predicted SDF on the surface? In my understanding, since the proposed method (almost) exactly finds the intersections of the hyperplanes, the predicted SDF on the entire reconstructed surface should be zero (or with a very small error from the approximations made when searching for the intersection). This is not the case for Marching Cubes which has a fixed grid over the domain.
> > >
> > > Plotting the predicted SDF value over the reconstructed surfaces should also give much better insights into where MC makes mistakes vs the proposed method and how severe they are.

---

> ### Author Response · Authors · 2024-08-11
>
> Regarding **Fig. A** (closer looks at bunny's nose), we understand you are referring to the small facets along the left-side of the nose line in the highlighted region. We assume these narrow facets are embedded in the facets of MC256, while MC64 fails to capture the left and right edge of the nose and represents this with more facets cutting these two sides of edges (*not aligned* with nose lines and rather assign more facets in here).
>
> Regarding the quantitative evaluation, we are happy to report the predicted SDF on the surface as you suggested. We randomly sampled 100K points on the surface of pretrained networks (a bunny Large model) and calculated the *average* of squares for predicted SDF (model's outputs) on the surface. As we expected, the predicted SDFs on the reconstructed surface converge to zero as the weight of the eikonal loss increases, and the number is close to zero $\approx 3e-8$. Note that, for a large model, fine-grained trilinear regions from the dense grid marks (from the more dense multi-resolution grids) allow relatively small errors of $97 \times 1e-9$ without the eikonal loss, although the eikonal loss of $1e-2$ further decreases to $31 \times 1e-9$.
>
> | Eikonal Loss Weight   |   Average of Squared SDFs |
> |-----------------------|----------------------- |
> | 0 | 97 x 1e-9 |
> | 1e-4 | 91 x 1e-9 |
> | 1e-3 | 68 x 1e-9 |
> | 1e-2 | 31 x 1e-9 |
>
> Plotting the predicted SDF values on the reconstructed surfaces would be beneficial. We could visualize this in a manner similar to **Fig. F**, which illustrated the normal errors on the surfaces. We anticipate that regions with high curvature will be particularly noteworthy. Notice that, sadly, we cannot update the rebuttal PDF during the author-reviewer discussion period.
>
> Apologies for overlooking your insights, and we hope this response addresses your concern.

---

> > ### Comment · Reviewer_MSAr · 2024-08-11
> >
> > Thank you a lot for such a fast response.
> >
> > Do you happen to also have these numbers for MC64 and MC256?

---

> ### Author Response · Authors · 2024-08-11
>
> We understand that, for the extracted mesh using MC64 or MC256, we can measure the average of predicted SDFs and how much they deviate from the learned surface of networks.
>
> In the same procedure, the corresponding MC numbers are as follows:
>
> | MC Sample | Average of Squared SDFs |
> |-|-|
> | 64 | 2308 x 1e-9 |
> | 256 | 66 x 1e-9 |
>
> We confirmed that the numbers are higher than the mesh using our method (31 x 1e-9).

---

> > ### Comment · Reviewer_MSAr · 2024-08-11
> >
> > Thank you.
> > I don't have any more questions and will adjust my rating. I think the paper is interesting and should be accepted, given the promised changes in the final version.

---

> > > ### Author Response · Authors · 2024-08-11
> > >
> > > Thank you for your thoughtful and detailed feedback. We truly appreciate the time you invested in reviewing our manuscript.

---

### Author Rebuttal · Authors · 2024-08-03

Please refer to the **rebuttal PDF (below PDF button)** attached for **Figures A-F** mentioned in the author's feedback. This PDF is high-resolution; please enlarge the PDF if you needed.

* **Fig. A.** Detailed mesh visualizations for Ours, MC, MT, and NDC.
* **Fig. B.** A visual explanation for a tropical polynomial and how a space is divided by it.
* **Fig. C.** A schematic diagram for 2D subdivision of our method.
* **Fig. D.** A plot of chamfer distances for Ours, MC, MT, and NDC.
* **Fig. E.** A plot of chamfer distances for Ours and MC applying QEM (gradual mesh simplification).
* **Fig. F.** Visualization of face normal errors on the surfaces.
----------------------------------------------------------------------
In summary, our work received the following positive reviews:

* **Motivation**:
  - "A very reasonable extension" (MSAr)
  - "Well motivated and holds a lot of promise," "first attempt" (FmNY)
* **Findings and Insights**:
  - For Thm 4.5 (Coro 4.6), "seem surprising," "great insight" (MSAr)
  - "Theoretical aspect of the work is in my view the most clear contribution" (FmNY)
  - "The theoretical framework is interesting and it is a good application of tropical algebra" (1oW4)
* **Results**: Proved mesh efficiency (MSAr, 1zQb)
* **Presentation**:
  - "Self-contained" (MSAr)
  - "Thorough theoretical analysis" (1zQb)
  - "Mathematical notation is good overall, with good choices for the symbols and their use" (1oW4)
----------------------------------------------------------------------
We recognize that, as an interdisciplinary work bridging algebraic geometry and neural 3D, we respect the raised issues and have carefully considered them, understanding the need for thorough analyses and clear explanations. We summarized our global feedback on the core raised issues:

**G1. Why does the chamfer efficiency matter?** (1zQb, FmNY)

- Although we found our method achieved the *lowest* chamfer distance in the Large models (see Tab. 6-7 in Appendix and **Fig. D**), we argued that the chamfer efficiency is *our highlight* since:
  1) sampling-based methods suffer exponential memory/compute costs to find optimal,
  2) for light-weight deploy of meshes (e.g., WebGL),
  3) and to get accurate normals for light modeling (see angular distance in Tab. 5 and **Fig. F**).

**G2. Additional Comparisons to Alternate Approaches (MT, NDC)** (MSAr, 1zQb, FmNY, 1oW4)

- Marching Cubes (MC) remains the de facto sampling-based method. In contrast, our method is a white-box method grounded in the theoretical understanding of trilinear interpolation and polyhedral complex derivation. Here is **our rationale for the evaluation**:

    1. Analytical approaches for CPWA functions are limited by their inability to effectively manage spectral bias and their exponential computational cost, which struggles with exponentially growing linear regions (infeasible to run our benchmarks).
    2. We also exclude optimization-based methods as they typically rely on MC or its variants initialization in their pipelines (our method can also be integrated into them), such as MeshSDF (Remelli et al., 2020), Deep Marching Tetrahedra (Shen et al., 2021), and FlexiCubes (Shen et al., 2023). Additionally, these methods result in prolonged computational costs.

- **MT & NDC results:** To be faithful to the reviewers' requests, we provide both **rigorous qualitative** (**Fig. A & F**) and **quantitative** (**Fig. D**) analyses on **Marching Tetrahedra (MT)** and **Neural Dual Contour (NDC, Chen et al., 2022)** in the rebuttal PDF. MT is a popular variant of MC that utilizes six tetrahedra within each grid to identify intermediate vertices. NDC is a data-driven approach (a pretrained model) that uses Dual Contour. In short, MT efficiently requires SDF values but is less efficient regarding the number of vertices extracted. (MT is better than MC in the same resolution, producing more vertices, though.) NDC faces a generalization issue for a zero-shot setting, producing unsmooth surfaces. We used the code from Deep Marching Tetrahedra (Shen et al., 2021) for MT and the official code of NDC. And PyMCubes for our *modernized* MC library.

**G3. Applying QEM** (1zQb, FmNY, 1oW4)

-  QEM is a popular mesh simplification algorithm, it could potentially improve both MC and our method for efficiency, which is validated in **Fig. E**. Overall, mesh simplification favors our method, as shown. Notice that QEM entails a higher computational cost, at least $\mathcal{O}(n \log n)$ (with $n$: the number of vertices). Additionally, these methods can cause shape deformation or holes, sensitive to hyperparameters. We used MeshLab's Quadric Edge Collapse Decimation with the default settings to halve the number of vertices for each step.

**G4. Better Presentation** (MSAr, 1oW4)

- As an interdisciplinary work, from tropical geometry to deep learning to implicit neural surface learning, we respond that the content needs visual aids to illustrate core concepts and provide clear explanations for inexperienced readers (1oW4). We are committed to making our paper more accessible. Specifically, we will: 1) simplify complex sentences and terminology where possible. 2) add a brief introductory section that explains Polyhedral Complex Extraction in simpler terms before delving into the technical details. For those who might miss it, please refer to Fig. 1, and Alg. 1 & 2 in the Appendix for visual and textual aids illustrating core concepts and procedures. Supplementarily, our new **Figure B & C (rebuttal PDF)**, B) a visual explanation for a tropical polynomial and how a space is divided by it, and C) a schematic diagram for 2D subdivision of our method would also help readers.

---

### Author Response · Authors · 2024-08-14
**Thanks and commitment to revisions**

As the author-reviewer discussion ends, we sincerely thank our reviewers for their thorough assessments and detailed, constructive feedback, which have been invaluable in improving our work. We are committed to revising our manuscript and appendix according to our discussions and will prioritize this.

Thank you once again for your dedication, time, and effort.

We truly appreciate the value of your contributions.


On behalf of all authors

---

### Decision · Program_Chairs · 2024-09-25

**Decision:**

Accept (poster)

**Comment:**

The paper proposes a new method for mesh extraction from piecewise trilinear networks, grounded in a theoretical grounding in trilinear interpolation and polyhedral complex derivation.  The method is shown to be more accurate and efficient in comparison to MC and other alternative approaches.

The paper received four reviews with unanimity in their (post discussion) ratings. The reviewers carefully engaged with the paper and the author response and participated in ensuing discussions. They found the paper to be well motivated, and a reasonable, theoretically grounded, extension of previous work on complex polyhedral complex extraction from ReLU networks to architectures with positional encodings implemented using trilinear interpolation. The reviewers particularly appreciated the theoretical grounding of the proposed approach, with interesting insights, and holding a lot of promise.

The initial reviews were mixed around the borderline. The primary concerns are around (a) the organization of the paper “focusing on lower abstraction details”, lacking supporting figures and visualizations compromising readability, and making the paper difficult to understand; (b) for a mathematical paper, riddled with typos, imprecise statements, poor phrasing, and undefined terms (definition before usage); (b) with unconvincing experimental evaluation – both qualitatively and quantitatively; inadequate comparison with alternate approaches, and, (d) an inadequate limitations section.

The authors addressed many of these concerns during the discussion resulting in two reviewers revising their ratings upward, resulting in a unanimous rating of ‘borderline accept’ from all reviewers. I recommend that the paper be accepted subject to the authors incorporating the clarifications and results from their rebuttal and responses, as well as other promised changes improving the presentation of the paper.